# FAIR ATTRIBUTE CLASSIFICATION VIA DISTANCE CO-VARIANCE

## ABSTRACT

With the increasing prevalence of machine learning, concerns about fairness have emerged. Mitigating potential discrimination risks and preventing machine learning algorithms from making unfair predictions are essential goals in fairness machine learning. We tackle this challenge from a statistical perspective, utilizing distance covariance—a powerful statistical method for measuring both linear and non-linear correlations—as a measure to assess the independence between predictions and sensitive attributes. To enhance fairness in classification, we integrate the sample distance covariance as a manageable penalty term into the machine learning process to promote independence. Additionally, we optimize this constrained problem using the Lagrangian dual method, offering a better trade-off between accuracy and fairness. Theoretically, we provide a proof for the convergence between sample and population distance covariance, establishing necessary guarantees for batch computations. Through experiments conducted on a range of real-world datasets, we demonstrate that our approach can seamlessly extend to existing machine learning models and deliver competitive results.

## 1 INTRODUCTION

Despite the success of modern deep neural networks (DNNs) applied to various tasks in recent years, it may raise many ethical and legal concerns during model training. In real-world classification and decision-making tasks, biased datasets can influence machine learning models, resulting in unfair predictions (Bellamy et al., 2018). With some online models and algorithms, unfair prediction results can lead users to make biased choices, and behavioral biases can create even more biased data, creating a cycle of bias (Mehrabi et al., 2021). Therefore, ensuring fairness is crucial when applying machine learning to tasks like computer vision, natural language processing, classification, and regression, requiring careful consideration of ethical and legal risks.

Fairness in machine learning can be broadly categorized into two levels: (1) group-level fairness, emphasizing equitable treatment among different groups, and (2) individual-level fairness, with the goal of providing similar predictions for similar individuals. In this paper, our focus is on group-level fairness. A fair machine learning model should avoid producing biased outputs based on sensitive attributes such as ethnicity, gender, and age. While these sensitive attributes may not be explicitly present in the training data features, deep learning models often work with high-dimensional and complex data, which contains a wealth of information. Some of this information may inadvertently correlate with sensitive attributes and result in biased outcomes (Kim et al., 2019; Park et al., 2021).

A natural and intuitive idea to enhance fairness is to reduce the reliance of prediction results on sensitive attributes. We treat the model's prediction outcomes and sensitive attributes as two random variables. If these two random variables are independent, it satisfies Demographic Parity (DP), a well-known fairness criterion. Measuring the independence of random variables is a fundamental objective in statistical data analysis, and there are various methods available for this purpose. Among these methods, Pearson correlation coefficients are commonly used to measure the correlation between random variables. This method has been extensively researched in the context of fair machine learning (Zafar et al., 2017; Zhao et al., 2020), yielding positive results and providing evidence for the effectiveness of this idea. However, it is important to note that Pearson correlation coefficients are primarily effective for capturing linear correlations and may struggle to capture non-linear

relationships. Moreover, the relationships between variables in complex datasets are often non-linear and can be further complicated by non-linear transformations performed by machine learning models.

Although Hirschfeld-Gebelein-Rényi (HGR) maximal correlation and mutual information (MI) are commonly used to assess the independence between two random variables or vectors, their exact computation is often infeasible. Instead, practitioners rely on estimations or upper/lower bounds to approximate independence. Due to the impossibility to traverse all functions, we can not compute the HGR maximal correlation accurately. As a result, classical methods have been developed to approximate HGR by constraining the traversed functions to linear spaces or Reproducing Kernel Hilbert Spaces Mary et al. (2019); Lee et al. (2022). Estimating and optimizing mutual information can also be challenging. To address these challenges, some methods replace MI with lower or upper bounds Song et al. (2019) or utilize variational methods Song & Ermon (2019) to approximate MI.

In this paper, we introduce a so-called distance covariance based method for fair classification. Distance covariance (DC) is a robust method for quantifying both linear and non-linear correlations between two random vectors (Székely et al., 2007). A smaller distance covariance value indicates a weaker relationship between the random variables, and it equals zero if and only if the two random variables are independent. However, directly computing distance covariance can be challenging as it requires knowledge of the analytical form of the distribution function and involves integration. To overcome this limitation, we employ empirical distance covariance, which can be computed directly from samples, as a substitute loss for fairness criteria. The discrepancy between the empirical DC and the population DC is attributed to **stochastic error**, which can be reduced by increasing the sample size. It is crucial to note that **the empirical DC almost surely converges to the population DC when the sample size tends to infinity.** For HGR and MI, they can use the estimations or upper/lower bounds instead. **Apart from stochastic error, there are also approximation errors** that arise due to the nature of the approximation itself. However, increasing the sample size does not necessarily drop the approximation error. In addition, the empirical DC is a continuously differentiable and biconvex function about the predicted target matrix and the sensitive attribute matrix. In the fairness classification problem, the sensitive attribute matrix is known, so the empirical DC is a continuously differentiable convex function of the predicted target matrix, which is an elegant property for optimization. We incorporate it as a regularization term during the model training process, utilizing the empirical DC between the predicted label (or feature map) and the sensitive attribute(s).

Although the empirical distance covariance converges almost surely to the population distance covariance, this property holds for large samples as the sample size approaches infinity. In deep learning, various batch gradient descent methods are commonly employed due to limitations on GPU memory. To address this, we provide a theoretical proof of the convergence in probability between the population and empirical distance covariance with respect to the sample size. This result offers crucial theoretical assurances for small-batch computations.

Moreover, considering the specific characteristics of fairness learning and distance covariance, we employ the Lagrangian dual method to address the constrained optimization problem. Manually determining the balance parameter for the fairness surrogate loss can be a challenging task. To tackle this issue, we propose treating the balance parameter as a dual variable and iteratively optimizing both the model parameters and the balance parameter. Numerical experimental results demonstrate that this approach enables the model to achieve a more favorable trade-off between accuracy and fairness.

Our method does not necessitate any prior knowledge about the model or existing bias, rendering it widely applicable. Furthermore, it is not restricted to binary sensitive attributes, but can be extended to encompass any number of sensitive attributes or subgroups. This adaptability stems from the fact that distance covariance can be applied to random vectors of any dimension. Moreover, our method is a plug-and-play approach, allowing for easy application across various domains, datasets, and neural network models with minimal computational cost. This characteristic enhances its practicality for real-world applications.

In summary, our contributions can be outlined as follows. (1) We introduce the utilization of empirical distance covariance as a feasible penalty term in the machine learning process to promote independence. Additionally, we employ the Lagrangian dual method to optimize this constrained problem, resulting in improved trade-offs between accuracy and fairness. (2) Instead of relying

on population distance covariance, we employ empirical distance covariance for calculations. The theoretical difference between the two, in terms of probability, is quantified and presented in Theorem 3. (3) The numerical experiments conducted in this paper provide evidence of the versatility of our proposed methods, showcasing their applicability across diverse datasets and tasks, while achieving competitive performance.

Throughout the paper, we use bold capital letter, capital letter to represent random vectors, sample matrices, respectively.

## 2 RELATED WORK

Fair machine learning is a significant field that has witnessed the advancement of various methods. In this section, we will primarily focus on the related work in fair classification. Existing fair machine learning methods can be broadly classified into three stages of processing: pre-processing (du Pin Calmon et al., 2017; Xu et al., 2018; Sattigeri et al., 2019; Ramaswamy et al., 2021), post-processing (Hardt et al., 2016; Bolukbasi et al., 2016; Pleiss et al., 2017; Mehrabi et al., 2022; Alghamdi et al., 2022), and in-processing (Zafar et al., 2017; Zhao et al., 2020; Mary et al., 2019; Lee et al., 2022; Moyer et al., 2018; Song et al., 2019; Creager et al., 2019; Chuang & Mroueh, 2020; Park et al., 2022; Liu et al., 2022; Guo et al., 2022; Lowy et al., 2021). Our method belongs to the in-processing category, where the concept of correlation is employed as a surrogate loss for fairness criteria. (Zafar et al., 2017; Zhao et al., 2020) use covariance as a fairness constraint to improve decision boundaries. Mary et al. (2019) and Lee et al. (2022) utilize the Hirschfeld-Gebelein-Renyi (HGR) maximal correlation to measure the dependence between the model output and sensitive attributes. They employ this measure as a penalty term within their frameworks to enforce fairness. However, it's worth noting that calculating the HGR maximal correlation can be challenging in computation. To mitigate this issue, both papers utilize alternative approaches such as kernel density estimate (KDE) or Soft-HGR as approximations to calculate the desired measure. From the perspective of information theory, mutual information(MI) and its variants can serve as proxies for fairness criteria, aiming to reduce mutual information to minimize the model's dependence on sensitive attributes (Moyer et al., 2018; Song et al., 2019; Creager et al., 2019). However, calculating mutual information can be challenging due to the lack of probability density function. Consequently, instead of directly estimating mutual information, the authors approximate it by their upper/lower bounds or some variational methods. Chuang & Mroueh (2020) regularize the models on paths of the mixup samples to ensure improved generalization in terms of both accuracy and fairness. Meanwhile, Park et al. (2022) propose a fair supervised contrastive loss to ensure fairness by penalizing the inclusion of sensitive attribute information for fair visual representation learning.

While distance covariance is not being introduced into deep learning for the first time, previous efforts have explored its application in contexts such as few-shot learning, interpretability, robust learning, and fairness. Xie et al. (2022) uses distance covariance to extract features after the backbone networks for few-shot classification, aiming to capture the dependence of these features and enhance the performance of few-shot classification models. Zhen et al. (2022) propose the utilization of partial distance correlation as an alternative to Canonical Correlation Analysis (CCA) for evaluating the correlation between feature spaces of different dimensions. Their approach involves employing distance covariance to elucidate the model's training process, disentangle representations, and enhance robustness. Specifically, they calculate the distance correlation of the residuals obtained from projecting the first two random variables onto the third variable. While this strategy shows promise in disentangling learned representations and improving model robustness, it may not be directly applicable to our fair classification tasks.

Both Liu et al. (2022) and Guo et al. (2022) have utilized distance covariance in fairness representation tasks. In their work, Liu et al. (2022) introduced population distance covariance as an alternative to mutual information (MI) and demonstrated its asymptotic equivalence to MI. However, computing either population distance covariance or MI requires knowledge of the analytical form of the respective distribution functions, which necessitates prior knowledge of the distribution function or density function. To address this, Liu et al. (2022) assumed that the random variables follow a multivariate normal distribution and employed the Variational Autoencoder technique proposed by Kingma & Welling (2013) to fit the normal distribution. Nevertheless, the assumption of a normal distribution can be overly restrictive and may not hold for real-world data, potentially leading to errors in the

analysis. Guo et al. (2022) propose a method for learning fair representations by incorporating graph Laplacian regularization. They explore the relationship between graph regularization and distance correlation, highlighting its significance in the context of fairness representation tasks. Similarly to the approach presented by Liu et al. (2022), Guo et al. (2022) employ a Variational Autoencoder (VAE) to learn representations, which relies on assumptions regarding the underlying distribution of the data.

## 3 METHOD

### 3.1 DISTANCE COVARIANCE

Let $\mathbf{Y} \in \mathbb{R}^p$ and $\mathbf{Z} \in \mathbb{R}^q$ be two continuous random vectors. Let $g_{\mathbf{Y},\mathbf{Z}}(y, z)$ be the joint probability density function, and $g_{\mathbf{Y}}(y), g_{\mathbf{Z}}(z)$ be marginal probability density functions. Let $f_{\mathbf{Y},\mathbf{Z}}(t, s) = \int e^{i(t^T y + s^T z)} g_{\mathbf{Y},\mathbf{Z}}(y, z) dy dz$ be the joint characteristic function, and $f_{\mathbf{Y}}(t) = \int e^{it^T y} g_{\mathbf{Y}}(y) dy$, $f_{\mathbf{Z}}(s) = \int e^{is^T z} g_{\mathbf{Z}}(z) dz$ be marginal characteristic functions corresponding to $\mathbf{Y}, \mathbf{Z}$. Distance covariance $\mathcal{V}(\mathbf{Y}, \mathbf{Z})$ is defined as $\mathcal{V}(\mathbf{Y}, \mathbf{Z}) = \iint_{\mathbb{R}^{p+q}} |f_{\mathbf{Y},\mathbf{Z}}(t, s) - f_{\mathbf{Y}}(t) f_{\mathbf{Z}}(s)|^2 w(t, s) \, dt ds$, where the weight $w(t, s) = (c_p c_q |t|^{1+p} |s|^{1+q})^{-1}$ with $c_d = \frac{\pi^{(1+d)/2}}{\Gamma((1+d)/2)}$ and $\Gamma$ being the gamma function. In practical scenarios, obtaining the probability density functions (cumulative distribution functions) can be challenging or even impossible. Consequently, directly acquiring the distance covariance becomes unfeasible. To address this issue, we resort to utilizing the empirical DC instead, which is guaranteed to be non-negative by Theorem 1 in (Székely et al., 2007).

Consider the observed samples $(Y_i, Z_i), i = 1, \ldots, n$, which are drawn from the joint distribution of random vectors $(\mathbf{Y}, \mathbf{Z}) \in \mathbb{R}^p \times \mathbb{R}^q$. Let $Y = [Y_1, \cdots, Y_n]$ and $Z = [Z_1, \cdots, Z_n]$ represent the two sample matrices. The empirical distance covariance (Székely et al., 2007) is then defined as follows:

**Definition 3.1** *The empirical distance covariance $\mathcal{V}_n(Y, Z)$ is the product of the corresponding double centered distance matrices of the samples: $\mathcal{V}_n(Y, Z) = \frac{1}{n^2} \sum_{k,l=1}^n A_{kl} B_{kl}$, where $A_{kl} = a_{kl} - \bar{a}_{k\cdot} - \bar{a}_{\cdot l} + \bar{a}_{\cdot\cdot}$, $B_{kl} = b_{kl} - \bar{b}_{k\cdot} - \bar{b}_{\cdot l} + \bar{b}_{\cdot\cdot}$, with $a_{kl} = \|Y_k - Y_l\|_2$, $\bar{a}_{k\cdot} = \frac{1}{n} \sum_{l=1}^n a_{kl}$, $\bar{a}_{\cdot l} = \frac{1}{n} \sum_{k=1}^n a_{kl}$, $\bar{a}_{\cdot\cdot} = \frac{1}{n^2} \sum_{l,k=1}^n a_{kl}$ and $b_{kl} = \|Z_k - Z_l\|_2$, $\bar{b}_{k\cdot} = \frac{1}{n} \sum_{l=1}^n b_{kl}$, $\bar{b}_{\cdot l} = \frac{1}{n} \sum_{k=1}^n b_{kl}$, $\bar{b}_{\cdot\cdot} = \frac{1}{n^2} \sum_{l,k=1}^n b_{kl}$.*

**Proposition 1 (Biconvex)** *Let $(Y_i, Z_i), i = 1, \ldots, n$ be the observed samples drawn from a joint distribution of $(\mathbf{Y}, \mathbf{Z})$. Denote $Y = [Y_1, \cdots, Y_n]$ and $Z = [Z_1, \cdots, Z_n]$. Then the empirical distance covariance $\mathcal{V}_n(Y, Z)$ is a biconvex function of $(Y, Z)$.*

**Proof.** We defer the proof to Proposition 5 in the appendix. ∎

It is worth to noting that $\mathcal{V}_n(Y, Z)$ is convex with respect to $Y$ when $Z$ is fixed, which is an elegant property for optimization.

### 3.2 THE FAIR CLASSIFICATION PROBLEM AND FAIR CLASSIFICATION MODEL

In the following we consider a standard fair supervised learning scenario. Let $\mathbf{X} \in \mathcal{X}$ be the predictive variables, $\mathbf{Y} \in \mathcal{Y} \subset \mathbb{R}^p$ be target variables or labels, $\mathbf{Z} \in \mathcal{Z} = \{Z_1, \cdots, Z_S\} \subset \mathbb{R}^q$ be a sensitive attribute. The datasets is a ternary pair of these variables $D = \{(x_i, y_i, z_i), i = 1, 2, \ldots, n\}$. Our aim is to find a fair model $\phi : \mathbf{X} \to \hat{\mathbf{Y}}$ with respect to the associated sensitive attribute. That is, $\hat{y}_i = \phi(x_i), \forall i \in [n]$. Therefore, it is crucial to establish a criterion for evaluating the model's performance from a fairness perspective.

While the fight against bias and discrimination has a long history in philosophy, psychology, and more recently in machine learning, there is still no universally accepted criterion for defining fairness due to cultural differences and varying preferences. In this context, we will focus on two fairness criteria: demographic parity (DP) (Dwork et al., 2012) and Equalized Odds (EO) (Hardt et al., 2016). The DP criterion requires that predictions $\hat{\mathbf{Y}}$ and sensitive attributes $\mathbf{Z}$ are independent, represented as $\hat{\mathbf{Y}} \perp \mathbf{Z}$. On the other hand, the EO criterion states that predictions $\hat{\mathbf{Y}}$ and sensitive attributes $\mathbf{Z}$ should be conditionally independent given the labels $\mathbf{Y}$, denoted as $\hat{\mathbf{Y}} \perp \mathbf{Z} | \mathbf{Y}$.

In this subsection, we focus on applying the demographic parity criterion to address the fair classification problem. Our objective is to discover a model $\phi_\theta : \mathbf{X} \mapsto \hat{\mathbf{Y}}$ that attains high classification accuracy while guaranteeing independence between the predicted outcomes $\hat{\mathbf{Y}}$ and the sensitive attribute $Z$, denoted as $\hat{\mathbf{Y}} \perp \mathbf{Z}$. Here, $\theta$ represents the network parameter. The selection of the network architecture is driven by the task at hand, with complex problems often requiring deep neural networks. The specific details regarding different tasks are deferred to Section 4.

Distance covariance, as a statistic for covariance, can also be used to characterize the independence between two random variables. For any random vectors $\mathbf{Y}$ and $\mathbf{Z}$, the value of $\mathcal{V}(\mathbf{Y}, \mathbf{Z})$ is zero if and only if $\mathbf{Y}$ and $\mathbf{Z}$ are independent (see Lemma 4). However, calculating the distance covariance requires knowledge of the probability density functions, which can be difficult or even impossible to obtain in practical scenarios. As a result, directly computing the distance covariance may not be feasible. To address this challenge, the empirical distance covariance is employed as a constraint term to ensure that the model satisfies the demographic parity criterion. The optimization model is

$$\min \ \mathcal{L}_{CE}(\hat{Y}, Y) \quad \text{s.t.} \ \mathcal{V}_n(\hat{Y}, Z) = 0, \tag{1}$$

where $\mathcal{V}_n(\hat{Y}, Z)$ is the empirical distance covariance. Then the corresponding Lagrangian function of (1) is $\mathcal{L}(\phi_\theta, \lambda) = \mathcal{L}_{CE}(\hat{Y}, Y) + \lambda \mathcal{V}_n(\hat{Y}, Z) = \mathcal{L}_{CE}(\phi_\theta(X), Y) + \lambda \mathcal{L}_{dc}(\phi_\theta(X), Z)$, where $\lambda > 0$ is the Lagrangian multiplier serving as a hyperparameter for balancing the fitting term and the distance covariance term. The hyperparameters play a crucial role in model optimization. However, determining the optimal hyperparameters manually can be challenging, especially without the insights gained from multiple experiments.

As discussed in proposition 1, empirical distance covariance is a biconvex function about two sample matrices. In the setting of our fair classification model, we use empirical distance covariance to depict the independence of the predicted label and sensitive attribute, where the sensitive attribute is from the data. Therefore, the empirical distance covariance exhibits convexity with respect to the predicted label. If we choose a model from the information autoencoding family (Zhao et al., 2018) or other models that optimize the predicted label in the distribution space (Song et al., 2019), the empirical distance covariance becomes a convex constraint, which is a desirable property for optimization.

Building on this inspiration, we consider $\lambda$ as a dual variable in this paper. Let $g(\lambda) = \inf_\theta \mathcal{L}(\phi_\theta, \lambda)$, the corresponding Lagrangian dual is

$$\max_\lambda g(\lambda) = \max_\lambda \inf_\theta \mathcal{L}(\phi_\theta, \lambda). \tag{2}$$

In fact, the model (2) provides an alternative if it is not easy to choose a specific $\lambda$ when we minimize $\mathcal{L}$ w.r.t. $\theta$. But model (2) becomes a max-min problem, which may be hard to solve directly. We update iteratively and the scheme is as follows: we give the value $\lambda_e$ and $\theta_e$, and find the optimal value of $\theta_{e+1}$ and $\lambda_{e+1}$ by

$$\theta_{e+1} = \arg\min_\theta \mathcal{L}(\phi_\theta, \lambda_e), \ \lambda_{e+1} = \lambda_e + \beta \cdot \frac{\partial \mathcal{L}(\phi_{\theta_e}, \lambda)}{\partial \lambda}\bigg|_{\lambda=\lambda_e} = \lambda_e + \beta \cdot \mathcal{L}_{dc}(\hat{Y}, Z), \tag{3}$$

where $\beta$ is the learning rate. In the optimization algorithm, we utilize a batch-wise approach by dividing the entire dataset into smaller batches. For each batch, we update the network parameters, which enables incremental improvements in the model's performance. Subsequently, after updating the parameters for all batches, we update the dual variable based on the average of the results obtained from each batch. Therefore, the whole algorithm can be summarized in Algorithm 1. Although the balancing parameter $\lambda$ is dynamically adjusted, the decision-maker also has some control over the fairness-utility trade-off curve by choosing the initial guess of $\lambda$ or other hyperparameters in the Primal Dual method.

In addition, fairness can also be based on the feature map. In this case, our objective is to find a feature map $\psi$ and a classifier $D$ such that the predicted outcome $\hat{Y} = D(\psi(X))$ approximates the true label $Y$, while maximizing the independence between the feature representation $\psi(X)$ and the sensitive attribute $Z$. To achieve this, we consider the cross-entropy loss as the optimization objective. The optimization model can be formulated as

$$\min \ \mathcal{L}_{CE}(\hat{Y}, Y) \quad \text{s.t.} \ \mathcal{V}_n(\psi(X), Z) = 0. \tag{4}$$

---

**Algorithm 1** The Lagrangian dual method with Distance Covariance

---

**Input:** Epoch $E$, Dataset $D = \{(x_i, y_i, z_i)\}_{i=1}^n$, inital guess $\lambda_0, \theta$, learning rates $\alpha, \beta$.

1: **for** $e = 0, 1 \ldots, E - 1$ **do**
2:      $l_{dc} = 0$,
3:      **for** MiniBatch $\mathcal{B} \in D$ **do**
4:          $\hat{Y}_{\mathcal{B}} = [\hat{y}_1, \cdots, \hat{y}_{|\mathcal{B}|}]^T$ with $\hat{y}_i = \phi_\theta(x_i)$;
5:          $l_{dc} \leftarrow l_{dc} + |\mathcal{B}| \cdot \mathcal{L}_{dc}(\hat{Y}_{\mathcal{B}}, Z_{\mathcal{B}})$,
6:          $\theta \leftarrow \theta - \alpha \left( \frac{1}{|\mathcal{B}|} \sum_{\mathcal{B}} \nabla_\theta \mathcal{L}_{CE}(\hat{y}_i, y_i, z_i) + \lambda_e \nabla_\theta \mathcal{L}_{dc}(\hat{Y}_{\mathcal{B}}, Z_{\mathcal{B}}) \right)$;
7:      $\lambda \leftarrow \lambda + \frac{\beta}{|D|} \cdot l_{dc}$.

---

Similarly, we can calculate the corresponding Lagrangian function and update the model parameter $\theta$ and dual variable $\lambda$ iteratively.

### 3.3 Rethinking Proposed Algorithm

In general, demographic parity (DP) and equalized odds (EO) are considered to be distinct fairness criteria because independence and conditional independence are not equivalent concepts. It is only possible to achieve both DP and EO when the sensitive attributes $\mathbf{Z}$ are independent of the labels $\mathbf{Y}$ (Barocas et al., 2017), which implies that the data distribution is the same across different sensitive attribute groups. As a result, many fairness methods are designed to address either DP or EO criteria specifically, or they require adjustments to accommodate different fairness criteria. It is important to carefully consider the specific fairness requirements and trade-offs when selecting and applying fairness methods in practice.

Note that in classification tasks, commonly used loss functions (such as cross-entropy and mean squared error) can be utilized to measure the closeness between the ground truth labels and the predicted labels. Therefore, (1) is equivalent to

$$\max_\theta P(\phi_\theta(X) = Y), \quad \text{s.t. } \phi_\theta(X) \perp Z. \tag{5}$$

The ideal case of (5) is $P(\phi_\theta(X) = Y) = 1$ and $\phi_\theta(X) \perp Z$. However, achieving both conditions simultaneously may not always be possible. Nevertheless, Model (5) suggests that the objective goes beyond achieving independence between the predictions $\phi_\theta(X)$ and the sensitive attribute $Z$. It also includes constraints on the predictive performance of the function $\phi_\theta : \mathbf{X} \mapsto \hat{\mathbf{Y}}$.

Intuitively, samples sharing the same sensitive attribute, regardless of their target classes, have a tendency to cluster together due to shared characteristics or patterns within those groups. Conversely, the fitting term related to the target attribute places greater emphasis on accurately classifying the majority group. This is because capturing the patterns and characteristics of the majority group is often more crucial for optimizing the model's overall performance. Suppose $Y = y$ is a majority class in the sensitive class $Z = Z_i$, but not a majority class in $Z = Z_j$. The worst case is $P(\hat{Y} = y | Y = y, Z = Z_i) = 1$ and $P(\hat{Y} = y | Y = y, Z = Z_j) = 0$ since $Y = y$ is not a majority class in $Z = A_j$, which implies strong dependence between $Y$ and $Z$. The introduction of independence seeks to break this dependence, leading to an increase in $P(\hat{Y} = y | Y = y, Z = Z_j)$, and resulting in a smaller EO value.

Due to space constraints, we have included numerical illustrations on the connection between DP and EO in the Appendix B.

### 3.4 Convergence Analysis

In this section, we discuss some theoretical properties of distance covariance and provide an estimation of the difference between the population distance covariance and empirical distance covariance in Theorem 3. First of all, we present the almost sure convergence of distance distance covariance.

**Lemma 2 (Theorem 2 in (Székely et al., 2007))** *Let* $\mathbf{Y} \in \mathbb{R}^p$ *and* $\mathbf{Z} \in \mathbb{R}^q$ *be two random vectors and* $Y, Z$ *be the corresponding sample matrices. If* $E\|\mathbf{Y}\|_2 < \infty$ *and* $E\|\mathbf{Z}\|_2 < \infty$, *then almost surely* $\lim_{n \to \infty} \mathcal{V}_n(Y, Z) = \mathcal{V}(\mathbf{Y}, \mathbf{Z})$.

The almost sure convergence in Lemma 2 is a strong convergence property, which implies both convergence in probability and convergence in distribution. This is a good enough property for a statistic in mathematical statistics. It ensures that as the sample size increases, the empirical estimate consistently approaches the true population value. It is worth noting that the almost sure convergence property discussed in Lemma 2 is contingent on having a sufficiently large sample size. This implies that a substantial number of samples is required to guarantee the convergence of the estimator.

In deep learning, working with a large number of samples is indeed crucial for effective model training. However, practical limitations, such as graphics card memory constraints, often restrict the amount of data that can be loaded at once. To overcome this challenge, mini-batch gradient descent is commonly employed, where only a subset of the training dataset (referred to as a mini-batch) is used to update the neural network parameters at each iteration.

When using empirical distance covariance as a substitute for distance covariance in such scenarios, there is no direct guidance on almost sure convergence. The reason is that the convergence properties of empirical estimators heavily rely on the sample size. In mini-batch training, the mini-batch size ($n_b$) represents the effective sample size used for parameter updates. As $n_b$ is small compared to the total dataset, the convergence behavior may differ from the traditional almost sure convergence. In the following, we will estimate the probability that the empirical distance covariance and the population distance covariance are sufficiently close in terms of sample size. Thus, it will be easy to control sample size while considering the error rate given.

The following theorem establishes the result of convergence in probability about the sample size.

**Theorem 3 (Convergence in probability)** *Let $\mathbf{Y} \in \mathbb{R}^p$ and $\mathbf{Z} \in \mathbb{R}^q$ be two sub-Gaussian random vectors and $Y = [Y_1, \cdots, Y_n]$, $Z = [Z_1, \cdots, Z_n]$ be the sample matrices. For any $\epsilon > 0$, there exist positive constants $C$ and $C_\epsilon$ such that*

$$P(|\mathcal{V}_n(Y, Z) - \mathcal{V}(\mathbf{Y}, \mathbf{Z})| > \epsilon) \leq \frac{C}{n} + 4nC_\epsilon \exp\left(-\sqrt{\frac{n}{\log n}}\right) + 2\exp(-Cn^2 \log n).$$

**Proof.** We defer the proof to Theorem 9 in the appendix. ∎

In our fair classification model, we consider $\mathbf{Y}$ and $\mathbf{Z}$ to be the random vectors related to the predicted label (or feature map) and the sensitive attribute, respectively. They are all bounded, otherwise the neural network will be unstable or not convergent. Note that bounded random vectors are all sub-Gaussian random vectors, so it is reasonable to assume $\mathbf{Y}, \mathbf{Z}$ to be sub-Gaussian random vectors.

## 4 Numerical Experiments

In this section, we present the numerical experiments results of our proposed method on four real-world datasets, including UCI Adult dataset, ACSIncome dataset and two image datasets. We compare our proposed method with the following baselines: **Vanilla**, **FairMixup** (Chuang & Mroueh, 2020), **HGR** (Mary et al., 2019), **FairDisCo** (Liu et al., 2022), **Dist-Fair** (Guo et al., 2022), **FSCL** (Park et al., 2022), **FERMI** Lowy et al. (2021), **FairProjection** Alghamdi et al. (2022). Since FSCL is primarily tailored for image datasets, when comparing FSCL to other methods applied to Tabular datasets, the data augmentation technique is employed as described in Gharibshah & Zhu (2022). The criteria used to assess the performance of fairness are $\Delta$DP and $\Delta$EO (Park et al., 2022):

$$\Delta\text{DP} = \frac{1}{\binom{S}{2}} \sum_{i<j} |P(\hat{\mathbf{Y}}|\mathbf{Z} = Z_i) - P(\hat{\mathbf{Y}}|\mathbf{Z} = Z_j)|,$$

$$\Delta\text{EO} = \frac{1}{2\binom{S}{2}} \sum_{y\in\{0,1\}} \sum_{i<j} |P(\hat{\mathbf{Y}} = y|\mathbf{Z} = Z_i, \mathbf{Y} = y) - P(\hat{\mathbf{Y}} = y|\mathbf{Z} = Z_j, \mathbf{Y} = y)|,$$

where $S$ represents the number of categories or groups within the sensitive attribute $\mathbf{Z}$. In the subsequent subsections, we present the results of our numerical experiments. For additional details and supplementary information, please refer to Appendix C.

## 4.1 TABULAR DATASET

In our study on tabular datasets, we utilized the widely recognized UCI Adult dataset (Dua & Graff, 2017) and ACSIncome dataset Ding et al. (2021).

For experiments, we use a four-layer Multilayer Perceptron (MLP) model with Rectified Linear Unit (ReLU) activation function and Cross Entropy Loss, referred to as the **Vanilla** model. All baselines use the same classifier network structure to ensure a fair comparison. The four subfigures in Figure 1 depict the trade-off curves between accuracy and either $\Delta$DP or $\Delta$EO for the UCI and ACSIncome datasets with different initial guesses of $\lambda$. In our experiments, the initial guesses we choose are in the interval $[1, 15]$. The results shown for accuracy and fairness criterion are averaged over 10 runs. In the second subfigure, non-uniform tick marks have been applied on the horizontal axis at both ends of the double slash. This adjustment aims to improve the visualization of the trade-off curve for better clarity and understanding.

In the figure, the position of the curve closer to the right indicates higher accuracy, while the curve positioned closer to the bottom signifies fairer outcomes according to the specified fairness metric. The results highlight the competitive performance of our method on the UCI dataset, as well as its ranking as the second best method on the ACSIncome dataset. This suggests that utilizing distance covariance as a regularizer yields superior performance in capturing independence, as our model achieves higher accuracy while maintaining the same level of $\Delta$DP.

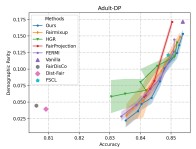 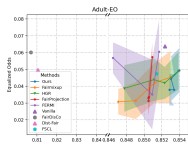 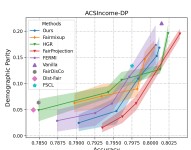 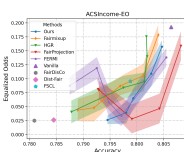

Figure 1: The results between accuracy and fairness for the UCI and ACSIncome datasets.

## 4.2 IMAGE DATA

In our experiments on image datasets, we conducted evaluations on both the CelebA and UTKFace datasets. For each classification task, we utilize the ResNet-18 architecture (He et al., 2016) as the encoder $\psi_\theta$ to process the input facial images. **Vanilla** refers to the model that utilizes the encoded representations obtained from $\psi_\theta$. These representations are then passed through a two-layer Multilayer Perceptron to make the final predictions. The predictions are based on the Cross Entropy loss function without any regularized term. All baselines share the same architecture as "Vanilla". The only exception is the Dist-fair model, which employs a VAE-ResNet18 architecture instead.

Due to the requirement of calculating the probability density function of the multivariate normal distribution in **FairDisCo** (Liu et al., 2022), its utilization becomes impractical when dealing with high-dimensional cases. Therefore, we are unable to offer comparison results for the image datasets as they typically possess extremely large dimensions. Furthermore, there may be a mistake in the implementation of the function *dis* within the provided code that computes the corresponding probability density function.

### 4.2.1 CELEBA

The CelebA dataset (Liu et al., 2015) is a widely used image dataset for various computer vision tasks, particularly those related to faces. In our experiments, we selected one or two sensitive attributes from the CelebA dataset to evaluate the performance of our method on mitigating potential ethical risks. For the classification tasks, we specifically focused on the attributes"attractiveness", "smile", "gender", and "wavy hair". These attributes were chosen based on two main criteria. First, we considered attributes that have a high Pearson correlation coefficient with the sensitive attributes. Second, we considered attributes where the subgroups based on the sensitive attributes are imbalanced. Imbalanced subgroups pose challenges for fairness evaluation, as the distribution of samples across different groups can impact the fairness of the classifier's predictions.

In Table 1, columns 2-5 present the classification accuracy and fairness (measured by $\Delta$DP or $\Delta$EO) for various combinations using one sensitive attribute. Due to the imbalanced nature of the CelebA

| Methods | Attractive/gender | | | Smile/gender | | | Wavy hair/gender | | | Attractive/young | | | Attractive/gender&young | | |
|---|---|---|---|---|---|---|---|---|---|---|---|---|---|---|---|
| | Acc | $\Delta$EO | $\Delta$DP | Acc | $\Delta$EO | $\Delta$DP | Acc | $\Delta$EO | $\Delta$DP | Acc | $\Delta$EO | $\Delta$DP | Acc | $\Delta$EO | $\Delta$DP |
| Vanilla | $82.188_{\pm0.4}$ | $20.514_{\pm0.7}$ | $39.722_{\pm3}$ | $92.379_{\pm0.2}$ | $3.160_{\pm0.7}$ | $14.384_{\pm1.3}$ | $83.147_{\pm0.4}$ | $19.229_{\pm1.6}$ | $30.96_{\pm1.7}$ | $82.129_{\pm0.2}$ | $22.912_{\pm0.4}$ | $46.072_{\pm0.5}$ | $82.149_{\pm0.2}$ | $24.418_{\pm0.6}$ | $34.649_{\pm0.3}$ |
| Ours | $80.568_{\pm0.3}$ | $\mathbf{0.563_{\pm0.3}}$ | $\mathbf{24.121_{\pm1}}$ | $\mathbf{92.751_{\pm0.4}}$ | $\mathbf{1.606_{\pm0.3}}$ | $10.336_{\pm1}$ | $81.785_{\pm0.2}$ | $\mathbf{1.122_{\pm0.6}}$ | $17.665_{\pm1.2}$ | $78.845_{\pm0.5}$ | $\mathbf{7.511_{\pm0.9}}$ | $34.371_{\pm0.6}$ | $77.532_{\pm1.0}$ | $\mathbf{8.107_{\pm1.0}}$ | $\mathbf{23.314_{\pm1}}$ |
| HGR | $80.423_{\pm0.2}$ | $1.175_{\pm0.2}$ | $24.720_{\pm1.5}$ | $92.620_{\pm0.2}$ | $2.055_{\pm0.2}$ | $15.213_{\pm0.5}$ | $80.139_{\pm0.7}$ | $2.052_{\pm0.4}$ | $17.156_{\pm1.2}$ | $78.796_{\pm0.7}$ | $8.74_{\pm0.8}$ | $31.91_{\pm0.9}$ | $78.320_{\pm0.2}$ | $9.571_{\pm0.4}$ | $26.149_{\pm0.2}$ |
| FSCL | $80.2_{\pm0.3}$ | $6.5_{\pm0.6}$ | $27.277_{\pm0.7}$ | $92.811_{\pm0.2}$ | $2.166_{\pm0.9}$ | $13.884_{\pm0.6}$ | $82.727_{\pm0.4}$ | $1.221_{\pm0.5}$ | $\mathbf{13.365_{\pm0.9}}$ | $81.425_{\pm0.2}$ | $16.125_{\pm1.2}$ | $40.632_{\pm1.5}$ | $78.945_{\pm0.5}$ | $17.636_{\pm1.6}$ | $30.341_{\pm1}$ |
| Dist-Fair | $70.738_{\pm1.2}$ | $13.108_{\pm3}$ | $26.169_{\pm3.8}$ | $74.227_{\pm0.6}$ | $2.707_{\pm0.3}$ | $\mathbf{10.084_{\pm0.5}}$ | $67.471_{\pm1}$ | $13.764_{\pm5.3}$ | $14.655_{\pm5.5}$ | $71.553_{\pm0.4}$ | $8.926_{\pm0.7}$ | $\mathbf{26.215_{\pm0.7}}$ | $71.0717_{\pm1.1}$ | $18.92_{\pm3.3}$ | $24.111_{\pm2.3}$ |
| Fair Mixup | $79.146_{\pm0.2}$ | $1.303_{\pm0.4}$ | $25.243_{\pm1}$ | $92.027_{\pm0.1}$ | $2.325_{\pm0.2}$ | $17.319_{\pm1.2}$ | $81.118_{\pm0.1}$ | $1.643_{\pm0.5}$ | $24.063_{\pm3.2}$ | $79.321_{\pm0.7}$ | $11.715_{\pm1.5}$ | $47.313_{\pm1.2}$ | — | — | — |

Table 1: Classification Results on CelebA.

dataset, it can be challenging to achieve both high accuracy and satisfy differential privacy (DP) simultaneously. Our method demonstrates strong competitiveness in achieving both high accuracy and fairness across these attribute combinations. The results indicate that our approach performs favorably compared to other methods in terms of balancing accuracy and fairness in classification tasks.

The last column of Table 1 demonstrates that our method maintains fairness in classification even in scenarios involving multiple sensitive attributes (Attractive/male & young). This ability is facilitated by the application of distance covariance, which allows for handling fair classification with arbitrary dimensional random vectors.

It is worth noting that FairMixup, another method employed in the evaluation, cannot be extended to fair classification with multiple sensitive attributes. Therefore, we have omitted their results in the last column of Table 1. This highlights the advantage of our method in addressing fairness concerns in classification tasks with multiple sensitive attributes.

### 4.2.2 UTKFACE

The UTKFace dataset (Zhang et al., 2017) is a dataset consisting of around 20,000 facial images. Each image is associated with attribute labels such as gender, age, and ethnicity. Following from Park et al. (2022), we consider "race" as the sensitive attribute and "gender" as the target attribute. During our evaluation, we divided the UTKFace dataset into training (80%) and testing (20%) sets.

To assess the impact of bias caused by the imbalance in the sensitive attribute (race), we introduced an imbalance factor denoted as $\alpha$. This factor determines the gender ratio within each sensitive group. Specifically, one sensitive group (e.g., Caucasian) has male data $\alpha$ times as much as female data, while the other sensitive group has the opposite gender ratio. More details about the data splittings with different imbalance factors can be found in Appendix C.4.

Table 2 presents the classification accuracy, $\Delta$EO, and $\Delta$DP results obtained from our evaluation. These results offer valuable insights into the impact of bias arising from the imbalance in the sensitive attribute, as well as highlight the effectiveness of our proposed method in mitigating such bias while simultaneously ensuring accuracy and fairness.

| Imbalance | $\alpha = 2$ | | | $\alpha = 3$ | | | $\alpha = 4$ | | |
|---|---|---|---|---|---|---|---|---|---|
| | Acc | $\Delta$EO | $\Delta$DP | Acc | $\Delta$EO | $\Delta$DP | Acc | $\Delta$EO | $\Delta$DP |
| Vanilla | $88.743_{\pm0.4}$ | $6.275_{\pm0.3}$ | $6.745_{\pm0.3}$ | $\mathbf{87.81_{\pm0.4}}$ | $10.454_{\pm0.4}$ | $10.676_{\pm0.3}$ | $85.246_{\pm1.0}$ | $12.940_{\pm0.6}$ | $13.319_{\pm0.1}$ |
| Ours | $\mathbf{89.198_{\pm0.4}}$ | $\mathbf{0.859_{\pm0.4}}$ | $\mathbf{0.918_{\pm0.4}}$ | $87.407_{\pm1.1}$ | $\mathbf{1.382_{\pm0.7}}$ | $\mathbf{1.349_{\pm0.7}}$ | $86.362_{\pm1.6}$ | $\mathbf{0.665_{\pm0.4}}$ | $\mathbf{0.726_{\pm0.3}}$ |
| HGR | $89.041_{\pm0.1}$ | $2.643_{\pm0.5}$ | $2.567_{\pm0.4}$ | $86.072_{\pm0.2}$ | $5.12_{\pm0.3}$ | $5.10_{\pm0.3}$ | $\mathbf{87.046_{\pm1.0}}$ | $8.077_{\pm1.3}$ | $8.113_{\pm1.4}$ |
| FSCL | $88.479_{\pm0.8}$ | $3.213_{\pm0.7}$ | $3.134_{\pm0.7}$ | $87.445_{\pm1.2}$ | $1.726_{\pm0.9}$ | $1.454_{\pm0.5}$ | $87.08_{\pm1.0}$ | $2.505_{\pm0.8}$ | $2.295_{\pm0.7}$ |
| Dist-Fair | $73.778_{\pm1}$ | $2.459_{\pm1.1}$ | $1.355_{\pm0.8}$ | $71.61567_{\pm2.0}$ | $3.538_{\pm1.1}$ | $2.424_{\pm2.1}$ | $73.159_{\pm0.4}$ | $5.218_{\pm0.5}$ | $4.967_{\pm0.5}$ |
| Fair Mixup | $87.613_{\pm0.3}$ | $2.643_{\pm0.5}$ | $2.123_{\pm1.2}$ | $85.565_{\pm0.5}$ | $2.105_{\pm0.1}$ | $1.849_{\pm0.3}$ | $86.55_{\pm1.0}$ | $1.329_{\pm0.6}$ | $1.015_{\pm0.3}$ |

Table 2: Classification Results On Imbalanced UTKFace.

Moreover, our method not only significantly enhances fairness but also improves classification accuracy compared to the vanilla model, particularly when the data imbalance is severe (e.g. $\alpha = 4$). This indicates that our approach not only rectifies fairness concerns but also leads to better overall performance in gender prediction tasks on imbalanced datasets.

## 5 CONCLUSION

In this paper, we introduces distance covariance as a powerful tool to depict fairness in machine learning, building on the independence between predictor and sensitive attribute. We analyze the properties of distance covariance and provide a convergence analysis of the empirical distance covariance in probability. To address the challenge of setting a balanced balance parameter, we

treat it as a dual variable and update it along with model parameters. Finally, we demonstrate the effectiveness and wide applicability of our proposed method through numerical experiments.

Future work in the field of fair machine learning encompasses several directions. While our method primarily focuses on classification tasks, it is essential to explore its applicability to regression problems with discrete/continuous sensitive attribute(s) as well. Furthermore, it is crucial to delve into the integration of statistical and optimization techniques with fairness in machine learning.

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

APPENDIX

## A  PROOFS

**Lemma 4 (Theorem 3 in (Székely et al., 2007))** *For any random variables $\mathbf{Y} \in \mathbb{R}^p$ and $\mathbf{Z} \in \mathbb{R}^q$ with $\mathbb{E}(\|\mathbf{Y}\|_2 + \|\mathbf{Z}\|_2) < +\infty$, we have $\mathcal{V}(\mathbf{Y}, \mathbf{Z}) = 0$ if and only if $\mathbf{Y}$ and $\mathbf{Z}$ are independent.*

**Proposition 5 (Proof of Proposition 1)** *Let $(Y_i, Z_i), i = 1, \ldots, n$ be the observed samples drawn from a joint distribution of $(\mathbf{Y}, \mathbf{Z})$. Denote $Y = [Y_1, \cdots, Y_n]$ and $Z = [Z_1, \cdots, Z_n]$. Then the empirical distance covariance $\mathcal{V}_n(Y, Z)$ is a biconvex function of $(Y, Z)$.*

**Proof.** noted that $Y_k - Y_l = [Y_1, Y_2, \ldots, Y_n][0, \ldots, 0, 1, 0, \ldots, 0, 1, 0, \ldots, 0]^T = YU_{kl}$ is a linear transformation of $X$. By Definition 3.1, we have

$$A_{kl} = a_{kl} - \bar{a}_{k\cdot} - \bar{a}_{\cdot l} + \bar{a}_{\cdot\cdot} = \|YU_{kl}\|_2 - \frac{1}{n}\sum_{l=1}^{n}\|YU_{kl}\|_2 - \frac{1}{n}\sum_{k=1}^{n}\|YU_{kl}\|_2 + \frac{1}{n^2}\sum_{l,k=1}^{n}\|YU_{kl}\|_2.$$

Since the $L_2$ norm is a convex function, $A_{kl}$ is also a convex function with respect to $Y$. Similarly, $B_{kl}$ is also a convex function with respect to $Z$. Thus, $\mathcal{V}_n(Y, Z) = \frac{1}{n^2}\sum_{k,l=1}^{n}A_{kl}B_{kl}$ is a convex function with respect to $Y$ when $Z$ is fixed and a convex function w.r.t. $Z$ when $Y$ is fixed. Therefore, the empirical distance covariance is a biconvex function of $(Y, Z)$. ∎

Before the proof of Theorem 3, we first introduce a lemma on the probability inequality of the $U$-statistic, from Theorem 5.6.1.A of Serfling (2009).

**Lemma 6** *Let $h(Y_1, \ldots, Y_m)$ be a kernel function of the $U$-statistic $U_n$, and $\theta = E\{h(Y_1, \ldots, Y_m)\}$. Let $a$ and $b$ be the upper and lower bounds of $h(Y_1, \ldots, Y_m)$. That is, $a \leq h(Y_1, \ldots, Y_m) \leq b$. For any $t > 0$ and $n > m$, we have*

$$P(U_n - \theta \geq t) \leq \exp\{-2[n/m]t^2/(b-a)^2\},$$

*where $[n/m]$ denotes the greatest integer function, i.e. the integer part of $n/m$.*

From Lemma 6 and the symmetry of $U$-statistic, we can get the following result:

$$P(|U_n - \theta| \geq t) \leq 2\exp\{-2[n/m]t^2/(b-a)^2\}, \tag{6}$$

which is the key point of our proof.

**Lemma 7** *For any $\epsilon > 0$, let $\hat{S}_1 = \frac{1}{n^2}\sum_{i,j=1}^{n}\|Y_i - Y_j\|_2\|Z_i - Z_j\|_2$ and $\tilde{S}_1 = \hat{S}_1\mathbb{I}(\hat{S}_1 \leq M_1)$, where $\mathbb{I}(\cdot)$ is an indicator function and $M_1 = \sqrt{\frac{n}{\log n}}\frac{\epsilon}{8}$ is a constant. Then there exists a constant $C > 0$ we have*

$$P(\{|\tilde{S}_1 - E(\tilde{S}_1)| > \frac{\epsilon}{4}\}) \leq \frac{C}{n}.$$

**Proof.** Let $h_1(Y_i, Y_j, Z_i, Z_j) = \|Y_i - Y_j\|_2\|Z_i - Z_j\|_2\mathbb{I}(\|Y_i - Y_j\|_2\|Z_i - Z_j\|_2 \leq M_1)$ be the kernel function. Then the corresponding $U$-statistic of the samples $Y_1, \ldots, Y_n$ with size $n$ is

$$\begin{aligned}
U_1 &= \frac{1}{n(n-1)}\sum_{i\neq j}h_1(Y_i, Y_j, Z_i, Z_j) \\
&= \frac{1}{n(n-1)}\sum_{i\neq j}\|Y_i - Y_j\|_2\|Z_i - Z_j\|_2\mathbb{I}\left(\|Y_i - Y_j\|_2\|Z_i - Z_j\|_2 \leq \sqrt{\frac{n}{\log n}}\frac{\epsilon}{8}\right).
\end{aligned} \tag{7}$$

Note that the kernel size $m = 2$, by Lemma 6, there exists a constant $C > 0$ we obtain

$$P(|U_1 - E(\tilde{S}_1)| \geq \epsilon) \leq 2\exp\left\{-2[n/m]\epsilon^2 / \left(\frac{n\epsilon^2}{64\log n}\right)\right\} = \frac{C}{n}. \tag{8}$$

Since

$$0 < U_1 - \tilde{S}_1 = \frac{1}{n(n-1)} \sum_{i \neq j} h_1(Y_i, Y_j, Z_i, Z_j) - \frac{1}{n^2} \sum_{i,j}^n h_1(Y_i, Y_j, Z_i, Z_j)$$

$$= \left( \frac{1}{n(n-1)} - \frac{1}{n^2} \right) \sum_{i \neq j} h_1(Y_i, Y_j, Z_i, Z_j) - \frac{1}{n^2} h_1(Y_i, Y_i, Z_i, Z_i)$$

$$= \left( \frac{1}{n(n-1)} - \frac{1}{n^2} \right) \sum_{i \neq j} h_1(Y_i, Y_j, Z_i, Z_j)$$

$$\leq \left( \frac{1}{n(n-1)} - \frac{1}{n^2} \right) (n(n-1))M = \frac{M}{n} = \frac{\epsilon}{8\sqrt{n \log n}} < \frac{\epsilon}{8},$$

when $n \geq 2$, so $P(\{|\tilde{S}_1 - U_1| > \frac{\epsilon}{8}\}) = 0$. Thus, from (8), get

$$P(\{|\tilde{S}_1 - E(\tilde{S}_1)| > \frac{\epsilon}{4}\}) = P(\{|\tilde{S}_1 - U_1 + U_1 - E(\tilde{S}_1)| > \frac{\epsilon}{4}\})$$

$$\leq P(\{|\tilde{S}_1 - U_1| > \frac{\epsilon}{8}\}) + P(\{|U_1 - E(\tilde{S}_1)| > \frac{\epsilon}{8}\}) \qquad (9)$$

$$\leq \frac{C}{n}.$$

∎

**Definition A.1 (Sub-Gaussian random variables, definition 5.7 in Vershynin (2010))** *A random variable $v$ is called a sub-Gaussian random variable if one of the following 4 conditions holds:*

- *Tails: the probability $P(|v| > t) \leq \exp(1 - t^2/K_1^2)$ for all $t \geq 0$;*

- *Moments: $(E|v|^p)^{1/p} \leq K_2 \sqrt{p}$ for all $p \geq 1$;*

- *Super-exponential moment: $E\left( \exp \frac{v^2}{K_3^2} \right) \leq e$;*

- *Moment generating function: $E \exp(tv) \leq \exp\left( \frac{K_4 t^2}{2} \right), \quad \forall\, t \in \mathbb{R}$.*

*where $K_i > 0, i = 1, 2, 3, 4$ are constants differing from each other by at most an absolute constant factor. The sub-Gaussian norm of $v$, denoted $\|v\|_{\psi_2}$, is defined to be*

$$\|v\|_{\psi_2} = \sup_{p \geq 1} p^{-1/2} (E|v - Ev|^p)^{1/p}.$$

In fact, the above 4 conditions are equivalent, see Lemma 5.5 of Vershynin (2010).

**Definition A.2 (Sub-Gaussian random vectors, definition 5.22 of Vershynin (2010))** *We say that a random vector $\boldsymbol{v} \in \mathbb{F}^d$ is sub-Gaussian if the one-dimensional marginals $\langle \boldsymbol{v}, \boldsymbol{u} \rangle$ are sub-Gaussian random variables for all $\boldsymbol{u} \in \mathbb{F}^d$. The sub-Gaussian norm of $\boldsymbol{v}$ is defined as*

$$\|\boldsymbol{v}\|_{\psi_2} = \sup_{\|\boldsymbol{u}\|=1} \|\langle \boldsymbol{v}, \boldsymbol{u} \rangle\|_{\psi_2}.$$

**Lemma 8** *Let $\mathbf{Y}, \mathbf{Z}$ be two sub-Gaussian random vectors. For any $t > 0$, there exists a constant $K_1 > 0$ such that*

$$P(\|\mathbf{Y}\|_2 > t) \leq 2 \exp(-t^2/K_1), \; P(\|\mathbf{Z}\|_2 > t) \leq 2 \exp(-t^2/K_1).$$

**Proof.** By the definition of sub-Gaussian random vector, we have $\|\mathbf{Y}\|_1$ is a sub-Gaussian random variable if $\mathbf{Y}$ is a sub-Gaussian random vector. Therefore, there exits a positive constant $K_1$ s.t.

$$P(\|\mathbf{Y}\|_1 > t) \leq 2 \exp(-t^2/K_1).$$

Note that $\|\mathbf{Y}\|_2 \leq \|\mathbf{Y}\|_1$, so $P(\|\mathbf{Y}\|_2 > t) \leq P(\|\mathbf{Y}\|_1 > t) \leq 2 \exp(-t^2/K_1)$. Similarly, we have $P(\|\mathbf{Z}\|_2 > t) \leq P(\|\mathbf{Z}\|_1 > t) \leq 2 \exp(-t^2/K_1)$ since $\mathbf{Z}$ is a sub-Gaussian random vector. ∎

**Theorem 9 (Proof of Theorem 3)** *Let $\mathbf{Y} \in \mathbb{R}^p$ and $\mathbf{Z} \in \mathbb{R}^q$ be two sub-Gaussian random vectors and $Y = [Y_1, \cdots, Y_n]$, $Z = [Z_1, \cdots, Z_n]$ be the sample matrices. For $\forall \epsilon > 0$, there exist positive constants $C$ and $C_\epsilon$ such that*

$$P(|\mathcal{V}_n(Y, Z) - \mathcal{V}(\mathbf{Y}, \mathbf{Z})| > \epsilon) \leq \frac{C}{n} + 4nC_\epsilon \exp\left(-\sqrt{\frac{n}{\log n}}\right) + 2\exp(-Cn^2 \log n).$$

**Proof.** Let $(\tilde{\mathbf{Y}}, \tilde{\mathbf{Z}})$ be an i.i.d copy of $(\mathbf{Y}, \mathbf{Z})$. By a simple calculation, we have
$$\mathcal{V}(\mathbf{Y}, \mathbf{Z}) = S_1 + S_2 - 2S_3,$$
where $S_i$, $i = 1, 2, 3$ are defined as follows:
$$S_1 = E\|\mathbf{Y} - \tilde{\mathbf{Y}}\|_2 \|\mathbf{Z} - \tilde{\mathbf{Z}}\|_2,$$
$$S_2 = E\|\mathbf{Y} - \tilde{\mathbf{Y}}\|_2 E\|\mathbf{Z} - \tilde{\mathbf{Z}}\|_2,$$
$$S_3 = E\left[E\|\mathbf{Y} - \tilde{\mathbf{Y}}\|_2 \big| \mathbf{Y}\right] E\left[E\|_2 \|\mathbf{Z} - \tilde{\mathbf{Z}}\|_2 \big| \mathbf{Z}\right].$$
The corresponding sample estimates are:
$$\mathcal{V}_n(Y, Z) = \hat{S}_1 + \hat{S}_2 - 2\hat{S}_3,$$
$$\hat{S}_1 = \frac{1}{n^2} \sum_{i,j=1}^n \|Y_i - Y_j\|_2 \|Z_i - Z_j\|_2,$$
$$\hat{S}_2 = \frac{1}{n^2} \sum_{i,j=1}^n \|Y_i - Y_j\|_2 \frac{1}{n^2} \sum_{i,j=1}^n \|Z_i - Z_j\|_2,$$
$$\hat{S}_3 = \frac{1}{n^3} \sum_{i,j,l=1}^n \|Y_i - Y_j\|_2 \|Z_i - Z_l\|_2.$$
Therefore,
$$P(|\mathcal{V}_n(Y, Z) - \mathcal{V}(\mathbf{Y}, \mathbf{Z})| > \epsilon) = P(\{|(\hat{S}_1 + \hat{S}_2 - 2\hat{S}_3) - (S_1 + S_2 - 2S_3)| > \epsilon\})$$
$$\leq P(\{|\hat{S}_1 - S_1| > \epsilon/4\}) + P(\{|\hat{S}_2 - S_2| > \epsilon/4\}) + P(\{|\hat{S}_3 - S_3| > \epsilon/4\}) \tag{10}$$
$$:= E_1 + E_2 + E_3,$$
where $E_1 = P(\{|\hat{S}_1 - S_1| > \epsilon/4\})$, $E_2 = P(\{|\hat{S}_2 - S_2| > \epsilon/4\})$, $E_3 = P(\{|\hat{S}_3 - S_3| > \epsilon/4\})$. In the following we will estimate the upper bounds of $E_1$, $E_2$ and $E_3$, respectively.

STEP 1.

We show the following statement. *For any $\epsilon > 0$, there exist constants $C, C_\epsilon > 0$ such that:* $P(\{|\hat{S}_1 - S_1| > \epsilon/4\}) \leq \frac{C}{n} + 4nC_\epsilon \exp\left(-\sqrt{\frac{n}{\log n}}\right)$.

Denote $\tilde{S}_1 = \hat{S}_1 \mathbb{I}(\hat{S}_1 \leq M_1)$, where $\mathbb{I}(\cdot)$ is an indicator function and $M_1 = \sqrt{\frac{n}{\log n}} \frac{\epsilon}{8}$ is a constant.

Define sets
$$G_1 = \{|\tilde{S}_1 - E(\hat{S}_1)| \leq \epsilon/4\},$$
$$G_2 = \{\tilde{S}_1 = \hat{S}_1\},$$
$$G_3 = \{\|Y_i - Y_j\|_2 \|Z_i - Z_j\|_2 \leq M_1, \ \forall i, j\},$$
$$G_4 = \{\|Y_i\|_2^2 + \|Z_i\|_2^2 \leq M_1/2, \ \forall i\},$$
$$G = \{|\hat{S}_1 - E(\hat{S}_1)| \leq \epsilon/4\},$$
we have
$$G_1 \cap G_2 \subset G, \ G_3 \subset G_2, \ G_4 \subseteq G_3,$$
where $G_4 \subseteq G_3$ can obtain since
$$\|Y_i - Y_j\|_2 \|Z_i - Z_j\|_2 \leq \frac{\|Y_i - Y_j\|_2^2 + \|Z_i - Z_j\|_2^2}{2}$$
$$= \frac{1}{2} \left(\|Y_i\|_2^2 + \|Y_j\|_2^2 - 2Y_i^T Y_j + \|Z_i\|_2^2 + \|Z_j\|_2^2 - 2Z_i^T Z_j\right)$$
$$\leq \left(\|Y_i\|_2^2 + \|Y_j\|_2^2 + \|Z_i\|_2^2 + \|Z_j\|_2^2\right) \leq 2\max_i \left(\|Y_i\|_2^2 + \|Z_i\|_2^2\right).$$

Thus, $G_3^c \subseteq G_4^c$ ($S^c$ denote the complementary set of the set $S$).

By the inclusion relation of the sets, we establish the following probability inequalities:

$$
\begin{aligned}
P(G^c) \leq P(G_1^c \cup G_2^c) &= P\left(G_1^c \cap (G_2 \cup G_2^c) \cup G_2^c\right) \\
&= P\left((G_1^c \cap G_2) \cup (G_1^c \cap G_2^c) \cup G_2^c\right) \\
&\leq P\left((G_1^c \cap G_2) \cup G_2^c\right) \\
&\leq P(G_1^c \cap G_2) + P(G_2^c) \\
&\leq P(G_1^c \cap G_2) + P(G_3^c) \\
&\leq P(G_1^c \cap G_2) + P(G_4^c).
\end{aligned}
\tag{11}
$$

By Lemma 7, there exists a constant $C > 0$, we have

$$
P(G_1^c \cap G_2) = P(\{|\tilde{S}_1 - E(\tilde{S}_1)| > \epsilon/4\}) \leq \frac{C}{n}.
\tag{12}
$$

Next, we estimate $P(G_4^c)$. There exists a constant $C_\epsilon > 0$, we have

$$
\begin{aligned}
P(G_4^c) &= P(\{\|Y_i\|_2^2 + \|Z_i\|_2^2 > M_1/2, \; \forall i\}) \\
&\leq P(\{\|Y_i\|_2^2 > \frac{M_1}{4}, \; \forall i\}) + P(\{\|Z_i\|_2^2 > \frac{M_1}{4}, \; \forall i\}) \\
&= P(\{\|Y_i\|_2 > \frac{\sqrt{M_1}}{2}, \; \forall i\}) + P(\{\|Z_i\|_2 > \frac{\sqrt{M_1}}{2}, \; \forall i\}) \\
&\leq 4n \exp\left(-\frac{M_1}{4K_1}\right) \leq 4nC_\epsilon \exp\left(-\sqrt{\frac{n}{\log n}}\right),
\end{aligned}
\tag{13}
$$

where the last inequality is based on Lemma 8, since $Y_i$, $Z_i$ are the Sub-Gaussian vectors.

Thus, combining (11), (12), (13), we have

$$
P(\{|\hat{S}_1 - S_1)| > \epsilon\}) = P(G^c) \leq P(G_1^c \cap G_2) + P(G_4^c) \leq \frac{C}{n} + 4nC_\epsilon \exp\left(-\sqrt{\frac{n}{\log n}}\right).
$$

STEP 2.

*For any $\epsilon > 0$, there exist constants $C, C_\epsilon > 0$ such that: $P(\{|\hat{S}_2 - S_2)| > \epsilon\}) \leq \frac{C}{n} + 2nC_\epsilon \exp\left(-\frac{n}{\log n}\right)$.*

Next, we consider $\hat{S}_2$. We write $\hat{S}_2$ as $\hat{S}_2 = \hat{S}_{2,1}\hat{S}_{2,2}$, where

$$
\hat{S}_{2,1} = \frac{1}{n^2}\sum_{i,j=1}^{n}\|Y_i - Y_j\|_2, \; \hat{S}_{2,2} = \frac{1}{n^2}\sum_{i,j=1}^{n}\|Z_i - Z_j\|_2
$$

Accordingly, $S_2 = S_{2,1}S_{2,2}$, where $S_{2,1} = E\|\mathbf{Y} - \tilde{\mathbf{Y}}\|_2, S_{2,2} = E\|\mathbf{Z} - \tilde{\mathbf{Z}}\|_2$.

Let $M_2 = \frac{\sqrt{n\epsilon}}{4\sqrt{\log n}}$. Choose the kernel functions $h_{2,1}(Y_i, Y_j) = \|Y_i - Y_j\|_2 \mathbb{I}(\|Y_i - Y_j\|_2 \leq M_2)$, $h_{2,2}(Z_i, Z_j) = \|Z_i - Z_j\|_2 \mathbb{I}(\|Z_i - Z_j\|_2 \leq M_2)$, let $\tilde{S}_{21} = \hat{S}_{21}\mathbb{I}(\hat{S}_{21} \leq M_2)$ and $\tilde{S}_{22} = \hat{S}_{22}\mathbb{I}(\hat{S}_{22} \leq M_2)$, and construct corresponding $U$-statistics:

$$
U_{2,1} = \frac{1}{n(n-1)}\sum_{i \neq j} h_{2,1}(Y_i, Y_j) = \frac{1}{n(n-1)}\sum_{i \neq j}\|Y_i - Y_j\|_2 \mathbb{I}(\|Y_i - Y_j\|_2 \leq M_2),
$$

$$
U_{2,2} = \frac{1}{n(n-1)}\sum_{i \neq j} h_{2,2}(Z_i, Z_j) = \frac{1}{n(n-1)}\sum_{i \neq j}\|Z_i - Z_j\|_2 \mathbb{I}(\|Z_i - Z_j\|_2 \leq M_2).
$$

Define sets

$$
\begin{aligned}
G_1 &= \{|\tilde{S}_{2,1} - E(\hat{S}_{2,1})| \leq \sqrt{\epsilon}/2\}, \\
G_2 &= \{\tilde{S}_{2,1} = \hat{S}_{2,1}\}, \\
G_3 &= \{\|Y_i - Y_j\|_2 \leq M_2, \; \forall i, j\}, \\
G_4 &= \{\|Y_i\|_2 \leq M_2/2, \; \forall i\}, \\
G &= \{|\hat{S}_{2,1} - E(\hat{S}_{2,1})| \leq \sqrt{\epsilon}/2\},
\end{aligned}
$$

$$P(G_1^c \cap G_2) = P(|\tilde{S}_{2,1} - E(\tilde{S}_{2,1})| > \sqrt{\epsilon}/2)$$
$$\leq P(|\tilde{S}_{2,1} - U_{2,1}| > \sqrt{\epsilon}/4) + P(|U_{2,1} - E(\tilde{S}_{2,1})| > \sqrt{\epsilon}/4)$$
$$\leq P(|U_{2,1} - E(\tilde{S}_{2,1})| > \sqrt{\epsilon}/4) \leq \frac{C}{n},$$

where the second inequality is from $|\tilde{S}_{2,1} - U_{2,1}| \leq \frac{M_2}{n^2(n-1)} \leq \sqrt{\epsilon}/4$ and the last inequality is by Lemma 6. Then we have

$$P(|\hat{S}_{2,1} - S_{2,1}| > \sqrt{\epsilon}/2) = P(G^c) \leq P(G_1^c \cap G_2) + P(G_4^c) \leq \frac{C}{n} + 2nC_\epsilon \exp\left(-\frac{n}{\log n}\right).$$

where $P(G_4^c) \leq P(\|Y_i\|_1 > \frac{M_2}{2}, \ \forall i) \leq 2nC_\epsilon \exp\left(-\frac{n}{\log n}\right)$ by Lemma 8. Similarly,

$$P(|\hat{S}_{2,2} - S_{2,2}| > \sqrt{\epsilon}/2) = P(G^c) \leq P(G_1^c \cap G_2) + P(G_4^c) \leq \frac{C}{n} + 2nC_\epsilon \exp\left(-\frac{n}{\log n}\right).$$

Since $\mathbf{Y}$ and $\mathbf{Z}$ are Sub-Gaussian random vectors, from the equivalent definition of Sub-Gaussian, we have with $K_1 > 0$

$$S_{2,1} = E\|\mathbf{Y} - \tilde{\mathbf{Y}}\|_2 \leq 2E\|\mathbf{Y}\|_2 \leq 2E\|\mathbf{Y}\|_1 \leq 4K_1,$$
$$S_{2,2} = E\|\mathbf{Z} - \tilde{\mathbf{Z}}\|_2 \leq 2E\|\mathbf{Z}\|_2 \leq 4K_1,$$

where we use the fact that the 1-norm of the sub-Gaussian random vector is a sub-Gaussian variable. Then, we can prove that:

$$P(|S_{2,1}(\hat{S}_{2,2} - S_{2,2})| > \epsilon/12) \leq P(|S_{2,1}||\hat{S}_{2,2} - S_{2,2}| > \epsilon/12)$$
$$\leq P(|\hat{S}_{2,2} - S_{2,2}| > \epsilon/(48K_1)) \leq \frac{C}{n} + 2nC_\epsilon \exp\left(-\frac{n}{\log n}\right),$$
$$P(|(\hat{S}_{2,1} - S_{2,1})S_{2,2}| > \epsilon/12) \leq P(|(\hat{S}_{2,1} - S_{2,1}||S_{2,2}| > \epsilon/12)$$
$$\leq P(|\hat{S}_{2,1} - S_{2,1}| > \epsilon/(48K_1)) \leq \frac{C}{n} + 2nC_\epsilon \exp\left(-\frac{n}{\log n}\right).$$
$$(14)$$

$$P(|(\hat{S}_{2,1} - S_{2,1})(\hat{S}_{2,2} - S_{2,2})| > \epsilon/12)$$
$$\leq P(|\hat{S}_{2,1} - S_{2,1}| > \sqrt{\epsilon/12}) + P(|\hat{S}_{2,2} - S_{2,2}| > \sqrt{\epsilon/12})$$
$$\leq \frac{C}{n} + 2nC_\epsilon \exp\left(-\frac{n}{\log n}\right).$$
$$(15)$$

Combining (14) and (15),

$$P(|\hat{S}_2 - S_2| > \epsilon/4) = P(|\hat{S}_{2,1}\hat{S}_{2,2} - S_{2,1}S_{2,2}| \geq \epsilon/4)$$
$$\leq P(|S_{2,1}(\hat{S}_{2,2} - S_{2,2})| > \epsilon/12) + P(|(\hat{S}_{2,1} - S_{2,1})S_{2,2}| > \epsilon/12)$$
$$+ P(|(\hat{S}_{2,1} - S_{2,1})(\hat{S}_{2,2} - S_{2,2})| > \epsilon/12)$$
$$\leq 3\left(\frac{C}{n} + 2nC_\epsilon \exp\left(-\frac{n}{\log n}\right)\right).$$
$$(16)$$

STEP 3.

*For any $\epsilon > 0$, there exist constants $C, C_\epsilon > 0$ such that: $P(\{|\hat{S}_3 - S_3)| > \epsilon\}) \leq \frac{C}{n} + 2\exp(-Cn^2 \log n) + 4nC_\epsilon \exp\left(-\sqrt{\frac{n}{\log n}}\right).$*

Denote $\tilde{S}_3 = \hat{S}_3 \mathbb{I}(\hat{S}_3 < M_3)$, where $\mathbb{I}(\cdot)$ is an indicator function and $M_3 = \sqrt{\frac{n}{\log n}} \frac{\epsilon}{48}$ is a constant.

Next, we construct the $U$-statistics related to $E(\tilde{S}_3)$ with the kernel function $h_3$,

$$\hat{h}_3(Y_i, Z_i, Y_j, Z_j, Y_l, Z_l) = \|Y_i - Y_j\|_2\|Z_i - Z_l\|_2 + \|Y_l - Y_j\|_2\|Z_i - Z_l\|_2$$
$$+ \|Y_i - Y_j\|_2\|Z_j - Z_l\|_2 + \|Y_i - Y_l\|_2\|Z_j - Z_l\|_2$$
$$+ \|Y_l - Y_j\|_2\|Z_i - Z_j\|_2 + \|Y_l - Y_i\|_2\|Z_i - Z_j\|_2,$$
$$h_3(Y_i, Z_i, Y_j, Z_j, Y_l, Z_l) = \hat{h}_3(Y_i, Z_i, Y_j, Z_j, Y_l, Z_l)\mathbb{I}(\hat{h}_3(Y_i, Z_i, Y_j, Z_j, Y_l, Z_l) < M),$$

$$U_3 = \frac{1}{\binom{n}{3}} \sum_{i<j<l} h_3(Y_i, Z_i, Y_j, Z_j, Y_l, Z_l).$$

Then we know by simple calculations

$$\tilde{S}_3 = \frac{(n-1)(n-2)}{6n^2}U_3 + \frac{n-1}{n^2}U_1,$$

where $U_1$ is defined in (7). By Boole's inequality, we have

$$
\begin{aligned}
P\left(\left|\tilde{S}_3 - E(\tilde{S}_3)\right| > \epsilon/4\right) &= P\left(\left|\frac{(n-1)(n-2)}{6n^2}U_3 + \frac{n-1}{n^2}U_1 - E(\tilde{S}_3)\right| > \epsilon/4\right)\\
&= P\Bigg(\left|\frac{(n-1)(n-2)}{n^2}\left(\frac{U_3}{6} - E(\tilde{S}_3)\right) + \frac{n-1}{n^2}\left(U_1 - E(\tilde{S}_1)\right)\right.\\
&\qquad \left. - \frac{3n-2}{n^2}E(\tilde{S}_3) + \frac{n-1}{n^2}E(\tilde{S}_1)\right| > \epsilon/4\Bigg)\\
&\leq P\left(\left|\frac{(n-1)(n-2)}{n^2}\left(\frac{U_3}{6} - E(\tilde{S}_3)\right)\right| > \epsilon/16\right) + P\left(\left|\frac{n-1}{n^2}\left(U_1 - E(\tilde{S}_1)\right)\right| > \epsilon/16\right)\\
&\quad + P\left(\frac{3n-2}{n^2}|E(\tilde{S}_3)| > \epsilon/16\right) + P\left(\frac{n-1}{n^2}|E(\tilde{S}_1)| > \epsilon/16\right).
\end{aligned}
\tag{17}
$$

For $E(\tilde{S}_3)$ and $E(\tilde{S}_1)$, since

$$\frac{n-1}{n^2}E(\tilde{S}_1) \leq \frac{n-1}{n^2}M_1 = \frac{\epsilon}{16\sqrt{n\log n}} < \frac{\epsilon}{16}, \tag{18}$$

and

$$\frac{3n-2}{n^2}E(\tilde{S}_3) < \frac{3n-2}{n^2}M_3 = \frac{(3n-2)\epsilon}{48n\sqrt{n\log n}} < \frac{\epsilon}{16}, \tag{19}$$

when $n > 2$, then $P(\{\|\frac{3n-2}{n^2}E(\tilde{S}_3)\|_2 > \epsilon/16\}) = 0$ and $P(\{\|\frac{n-1}{n^2}(U_1 - E(\tilde{S}_1))\|_2 > \epsilon/16\}) = 0$.
Note that

$$
\begin{aligned}
P&\left(\left|\frac{(n-1)(n-2)}{n^2}\left(\frac{U_3}{6} - E(\tilde{S}_3)\right)\right| > \epsilon/16\right)\\
&= P\left(\left|\left(\frac{U_3}{6} - E(\tilde{S}_3)\right)\right| > \epsilon/16\frac{n^2}{(n-1)(n-2)}\right)\\
&\leq P\left(\left|\left(\frac{U_3}{6} - E(\tilde{S}_3)\right)\right| > \epsilon/16\right) \leq \frac{C}{n},
\end{aligned}
\tag{20}
$$

where the last inequality is by Lemma 6. Similarly,

$$P\left(\left|\frac{n-1}{n^2}\left(U_1 - E(\tilde{S}_1)\right)\right| > \epsilon/16\right) \leq 2\exp(-Cn^2\log n). \tag{21}$$

Combining (18), (19), (21) and (20), (17) can be rewritten as

$$P\left(\left|\tilde{S}_3 - E(\tilde{S}_3)\right| > \epsilon/4\right) \leq \frac{C}{n} + 2\exp(-Cn^2\log n).$$

Define sets:

$$
\begin{aligned}
H_1 &= \{|\tilde{S}_3 - E(\hat{S}_3)| \leq \epsilon/4\},\\
H_2 &= \{\tilde{S}_3 = \hat{S}_3\},\\
H_3 &= \{\hat{h}_3(Y_i, Z_i, Y_j, Z_j, Y_l, Z_l) < M, \ \forall i, j, l\},\\
H_4 &= \{\|Y_i\|_2^2 + \|Z_i\|_2^2 \leq M/6, \ \forall i\},\\
H &= \{|\hat{S}_3 - E(\hat{S}_3)| \leq \epsilon/4\}.
\end{aligned}
$$

Similar with (11), we have $P(H^c) \leq P(H_1^c \cap H_2) + P(H_4^c)$. Thus,

$$
\begin{aligned}
&P(\{|\hat{S}_3 - S_3)| > \epsilon/4\}) = P(\{|\hat{S}_3 - E(\hat{S}_3)| > \epsilon/4\}) \leq P(H_1^c \cap H_2) + P(H_4^c) \\
&\leq P(\{|\tilde{S}_3 - E(\tilde{S}_3)| > \epsilon/4\}) + P(\{\|Y_i\|_2^2 + \|Z_i\|_2^2 > M_3/6, \ \forall i\}) \\
&\leq P(\{|\tilde{S}_3 - E(\tilde{S}_3)| > \epsilon/4\}) + P\left(\{\|Y_i\|_2 > \frac{\sqrt{M_3}}{\sqrt{12}}, \ \forall i\}\right) + P\left(\{\|Z_i\|_2 > \frac{\sqrt{M_3}}{\sqrt{12}}, \ \forall i\}\right) \\
&\leq \frac{C}{n} + 2\exp(-Cn^2 \log n) + 4nC_\epsilon \exp\left(-\sqrt{\frac{n}{\log n}}\right).
\end{aligned}
$$

In summary, we conclude the result. ∎

## B EXPLORING THE POTENTIAL RELATIONSHIP BETWEEN DP, EO AND ACCURACY

In Subsection 3.3, we try to think about the connections between DP and EO and find:

Model $\max_\theta P(\phi_\theta(X) = Y), \text{s.t. } \phi_\theta(X) \perp Z$ suggests that the objective goes beyond achieving independence between the predictions $\phi_\theta(X)$ and the sensitive attribute $Z$.

Intuitively, samples sharing the same sensitive attribute, regardless of their target classes, have a tendency to cluster together due to shared characteristics or patterns within those groups. Conversely, the fitting term related to the target attribute places greater emphasis on accurately classifying the majority group. This is because capturing the patterns and characteristics of the majority group is often more crucial for optimizing the model's overall performance.

Suppose $Y = y$ is a majority class in the sensitive class $Z = Z_i$, but not a majority class in $Z = Z_j$. The worst case is $P(\hat{Y} = y | Y = y, Z = Z_i) = 1$ and $P(\hat{Y} = y | Y = y, Z = Z_j) = 0$ since $Y = y$ is not a majority class in $Z = A_j$, which implies strong dependence between $Y$ and $Z$. The introduction of independence seeks to break this dependence, leading to an increase in $P(\hat{Y} = y | Y = y, Z = Z_j)$, and resulting in a smaller EO value.

To further investigate the relationships between DP, EO, and Accuracy, we conducted experiments on the CelebA and UTKFace datasets.

For CelebA dataset, our experiments focus on the "attractive" attribute as the target attribute and the "gender" attribute as the sensitive attribute. To analyze the impact of data fitting term on the DP and EO, we utilized a fixed balancing parameter $\lambda$ during training. In Figure 2, we present the model's performance on the test set with different $\lambda$. As the $\lambda$ parameter raising, both Acc and $\Delta$DP decrease. Conversely, $\Delta$EO demonstrates a $V$-shaped or increasing trend. Therefore, we may numerically choose an appropriate parameter such DP and EO are smaller with a high predicted accuracy.

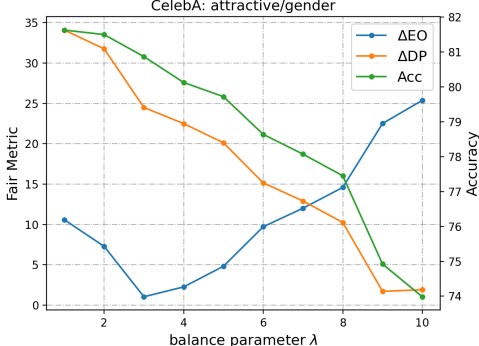

Figure 2: Trend between accuracy and fairness metrics for CelebA dataset with different balanced parameters.

We also provide detailed insights into the UTKFace dataset. Despite intentionally inducing imbalances in the dataset, we observed a simultaneous decrease in both the differential privacy (DP) and equal opportunity (EO) metrics alongside an increase in accuracy as the number of epochs progressed. Figure 3 illustrates the relationship between accuracy and fairness metrics (DP and EO) for the UTKFace dataset, specifically with an imbalance factor of $\lambda = 2$.

From the figure, we can observe that accuracy, $\Delta$EO, and $\Delta$DP tend to stabilize on this dataset when epoch number is larger than 40. Additionally, $\Delta$EO and $\Delta$DP exhibit relatively small differences and can be simultaneously reduced.

In our setting for each sensitive attribute class, the ratio of the majority class to the minority class in relation to the target classes is the same. This observation can possibly be attributed to the fact that we maintained a consistent imbalance factor across all sensitive attribute classes.

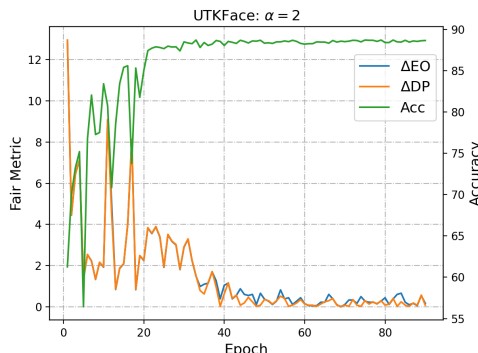

Figure 3: Trend between accuracy and fairness metrics for UTKFace dataset as the epoch increases.

## C  ADDITIONAL DETAILS ABOUT NUMERICAL RESULTS

The experiments are conducted in a Linux environment using the PyTorch library, utilizing the computational capabilities of a NVIDIA A100 Tensor Core GPU. In the classification task, the cross-entropy loss function is employed. A batch size of 256 is used during training, and the initial learning rate is set to 0.1.

### C.1  ADULT

We follow the preprocessing procedures outlined as Yurochkin et al. (2019); Chuang & Mroueh (2020). During the training process, Stochastic Gradient Descent (SGD) with momentum is utilized as the optimization algorithm. The model is trained for a total of 40 epochs, with the learning rate decayed by a factor of 10 at the 15th and 30th epochs.

In our experiments, we employ a dataset splitting strategy that involved dividing the data into training, validation, and testing sets. The proportions used for the splits are 60% for training, 20% for validation, and 20% for testing. To address the potential variability introduced by different dataset splits, we apply 10 different random seeds for the splitting process. For each split, we conduct the experiments and record the results. The reported results are the average over multiple experiments.

### C.2  ACSINCOME

ACSIncome is a dataset to predict whether an individual's income is above \$50,000, after filtering the 2018 US-wide ACS PUMS data sample to only include individuals above the age of 16, who reported usual working hours of at least 1 hour per week in the past year, and an income of at least \$100. When loading dataset, we follow the instruction in their *readme* file [1]. There are $22,268$ samples with 10 features.

---

[1] https://github.com/socialfoundations/folktables/tree/main

In the numerical experiments, we consider "Race" as the sensitive attribute with 2 classes: while and others. We followed the same approach as in the Adult dataset for dataset splitting and result recording.

## C.3 CELEBA

The CelebA dataset consists of over 200,000 celebrity images, with annotations for various attributes such as gender, age, and presence of facial hair. The CelebA dataset is known for its imbalanced class distribution, particularly with attributes such as gender, where the number of male and female samples is significantly different. This dataset presents a challenge for fairness evaluation due to the inherent bias in the data.

In our experiments on the CelebA dataset, the dataset was divided into training, validation, and testing sets, with sizes of $162k$, $18k$, and $19k$ samples, respectively. To encode the input data and extract meaningful representations, we utilized the ResNet-18 architecture. Following the encoding step, we employed a two-layer neural network for prediction. The neural network consisted of a hidden layer with a size of 100 neurons, and the Rectified Linear Unit (ReLU) activation function was applied.

During the training process, we utilized Stochastic Gradient Descent (SGD) with momentum as the optimization algorithm. The model was trained for a total of 90 epochs. The learning rate was decayed by a factor of 10 at the 20th, 40th, and 60th epochs. This learning rate decay strategy helps facilitate convergence and improve the model's performance over the course of training.

In this paper, we focus on analyzing four groups of training and test datasets that contain binary sensitive attributes. Each group consists of a total of 162,770 training samples and 19,962 test samples.

To provide detailed insights into the dataset characteristics, we present the sample numbers and proportions for different target/sensitive attribute settings in Table 3. Additionally, Table 4 provides the sample numbers and proportions for different target/sensitive attribute settings, specifically focusing on scenarios involving two sensitive attributes.

| Training | 162770 | | 162770 | | 162770 | | 162770 | |
|---|---|---|---|---|---|---|---|---|
| | Attractive/gender | | Smile/gender | | Wavy hair/gender | | Attractive/young | |
| | Target c0 | Target c1 | Target c0 | Target c1 | Target c0 | Target c1 | Target c0 | Target c1 |
| Sensitive Class 0 | 29920 (18.38%) | 49247 (30.26%) | 43688(26.84%) | 41002 (25.19%) | 52289 (32.12%) | 58499 (35.94%) | 30618 (18.81%) | 48549 (29.83%) |
| Sensitive Class 1 | 64589 (39.68%) | 19014 (11.68%) | 50821 (31.22%) | 27259 (16.75%) | 42220 (25.94%) | 9762 (6.00%) | 5364 (3.30%) | 78239 (48.07%) |
| Testing | 19962 | | 19962 | | 19962 | | 19962 | |
| | Attractive/gender | | Smile/gender | | Wavy hair/gender | | Attractive/young | |
| | Target c0 | Target c1 | Target c0 | Target c1 | Target c0 | Target c1 | Target c0 | Target c1 |
| Sensitive Class 0 | 4263 (21.36%) | 5801 (29.06%) | 5354(26.82%) | 4621 (23.15%) | 6178(30.95%) | 6517 (32.65%) | 4066 (20.37%) | 5998 (30.05%) |
| Sensitive Class 1 | 7984(40.00%) | 1914 (9.59%) | 6893(34.53%) | 3094 (15.50%) | 6069(30.40%) | 1198 (6.00%) | 782 (3.92%) | 9116 (45.67%) |

Table 3: Compositions of the CelebA datasets with a binary sensitive attribute.

| Traning | | Sensitive Attribute | | | |
|---|---|---|---|---|---|
| | Target Attribute | Female and Old | Female and Young | Male and Old | Male and Young |
| | Attractive | 7522 (4.62%) | 23096 (14.19%) | 22398 (13.76%) | 26151 (16.07%) |
| | Unattractive | 3645 (2.24%) | 1719 (1.06%) | 60944 (37.44%) | 17295 (10.63%) |
| Testing | Attractive | 1299 (6.51%) | 2767 (13.86%) | 2964 (14.85%) | 3034 (15.20%) |
| | Unattractive | 617 (3.09%) | 165 (0.83%) | 7367 (36.91%) | 1749 (8.76%) |

Table 4: Compositions of the CelebA datasets with multiple sensitive attributes.

## C.4 UTKFACE

In our approach, we utilize the ResNet-18 architecture to encode the input data into a representation of dimension 100. After encoding the input data, we employ a two-layer neural network for prediction. This neural network consists of a hidden layer with a size of 100 neurons and applies the Rectified Linear Unit (ReLU) activation function. The training model we use is the cross-entropy loss.

During the training process, we employ Stochastic Gradient Descent (SGD) with momentum as the optimization algorithm. We train the model for a total of 90 epochs. To facilitate effective training, we implement a learning rate decay strategy. Specifically, we reduce the learning rate by a factor of ten at the 20th, 40th, and 60th epochs.

In this dataset, we examine the target attribute of gender, which includes two categories: Male and Female. The sensitive attribute we focus on is ethnicity, specifically distinguishing between Caucasian and non-Caucasian groups. Table 5 outlines the sample numbers and proportions based on different data imbalance settings, where $\alpha$ is the imbalance factor.

| Imbalance($\alpha$) | 2 | | 3 | | 4 | |
|---|---|---|---|---|---|---|
| Training | 11972 | | 10716 | | 10022 | |
| | Ethnicity | | | | | |
| Gender | Caucasian | Others | Caucasian | Others | Caucasian | Others |
| Male | 3991 (33.34%) | 1995 (16.66%) | 4019 (37.50%) | 1339 (12.50%) | 4009 (40%) | 1002 (10%) |
| Female | 1995 (16.66%) | 3991 (33.34%) | 1339 (12.50%) | 4019 (37.50%) | 1002 (10%) | 4009 (40%) |
| Testing | 3664 | | 3564 | | 3704 | |
| | Ethnicity | | | | | |
| Gender | Caucasian | Others | Caucasian | Others | Caucasian | Others |
| Male | 916 (25.00%) | 916 (25.00%) | 1782 (50.00%) | 1782 (50.00%) | 1852 (50.00%) | 1852 (50.00%) |
| Female | 916 (25.00%) | 916 (25.00%) | 1782 (50.00%) | 1782 (50.00%) | 1852 (50.00%) | 1852 (50.00%) |

Table 5: Composition of the UTKFace datasets.

