# Rebuttal For ICLR2024

## 1 Rebuttal for Reviewer 8JtY

### 1.1 Weakness

1. The intuitive explanation for the distance covariance and its existing usages in statistics are not well explained.

**Answer:** Thanks for your comment. In the submitted version, the related works section highlights the computational disadvantages of Hirschfeld-Gebelein-Rényi (HGR) maximal correlation and mutual information (MI). In this revised version, we add discussions about advantages and disadvantages of the distance covariance, HGR and MI in the introduction section.

Two random vectors (variables) are independent if and only if any of their HGR maximal correlation, MI, and distance covariance (DC) obtain a value of 0. Although all the above three can be used to characterize the independence of two random variables (vectors), the DC stands out in terms of its computational efficiency and suitability for optimization.

The definition of HGR of two random vectors is

$$HGR(U, V) = \sup_{f,g} \langle f(U), g(V) \rangle.$$

In practical computations, it is IMPOSSIBLE to traverse all $f, g$. Some classical methods are to approximate HGR by requiring $f, g$ belonging to linear space or Reproducing Kernel Hilbert Spaces. In [1], the authors solve it through a kernel approximation.

MI terms are difficult to estimate and optimize [2]. In [2], they replace both mutual information of the objective function and the function in constraint by their lower bound and upper bounds respectively.

For DC,

1. Empirical DC, as a statistic, can be directly computed from the samples, although population DC [3] is also challenge for computing since it requires knowledge of the analytical form of the distribution function and involves integration.

2. The empirical DC of the predicted target and sensitive attribute is continuously differentiable and biconvex when we treat the predicted target as a free variable. Furthermore, it can be expressed in matrix form, which makes it relatively more suitable for optimization and efficient for computation.

[1]. J. Mary, C. Calauzènes, and N. E. Karoui. Fairness-Aware Learning for Continuous Attributes and Treatments. ICML, 4382–4391, 2019.

[2]. J. Song, P. Kalluri, A. Grover, S. Zhao, and S. Ermon. Learning Controllable Fair Representations. AISTAT, 2164–2173, 2019.

[3]. J. Liu, Z. Li, Y. Yao, F. Xu, X. Ma, M. Xu, and H. Tong. Fair Representation Learning: An Alternative to Mutual Information. SIGKDD, 1088–1097, 2022.

2. There are some missing references for fairness interventions, which I encourage the authors to include and compare, for example,

a. Lowy, A., Baharlouei, S., Pavan, R., Razaviyayn, M. and Beirami, A., 2021. A stochastic optimization framework for fair risk minimization. arXiv preprint arXiv:2102.12586.

b. Alghamdi, W., Hsu, H., Jeong, H., Wang, H., Michalak, P., Asoodeh, S. and Calmon, F., 2022. Beyond Adult and COMPAS: Fair multi-class prediction via

information projection. Advances in Neural Information Processing Systems, 35, pp.38747-38760.

**Answer**: Thanks for your recommendation, we add them to related works and numerical experiment parts.

3. Despite that the distance covariance is interesting, the reason why it is potentially a better metrics than other information-theoretic quantities such as mutual information and the Renyi maximal correlation is unclear to me.

Distance covariance, MI and the maximal correlation are all zero when two random variables are independent; however, the maximal correlation satisfies Renyi's postulates for a good measure of dependency.

It is encouraged that the authors spend more space to discuss the pros and cons regarding the dependency metrics, give illustrations on why one is better than the other and hopefully provide a simple numerical example.

**Answer:** Thanks for your comment, we add this part to the introduction section.

Although all the above three can be used to characterize the independence of two random variables (vectors), the distance covariance (DC) stands out in terms of its computational efficiency and suitability for optimization.

For DC, Empirical DC, as a statistic, can be directly computed from the samples, although population DC [1] is also challenge for computing since it requires knowledge of the analytical form of the distribution function and involves integration.

In addition, the empirical DC is a continuously differentiable and biconvex function about the predicted target matrix and the sensitive attribute matrix. In the fairness classification problem, the sensitive attribute matrix is known, so the empirical DC is a continuously differentiable convex function of the predicted target matrix, which is an elegant property for optimization.

Moreover, the paper [1] also demonstrates the advantages of DC by comparing with methods based on mutual information [2,3], demonstrating its advantages. We further utilized empirical distance covariance, not only reducing training time but also improving both accuracy and fairness.

For HGR and MI, we can only use the estimations or upper/lower bounds instead, which are used to approximate independence. In practice, this inevitably introduces biases and errors, subsequently reducing the utility of downstream classification tasks.

The definition of HGR of two random vectors is

$$HGR(U,V) = \sup_{f,g} \langle f(U), g(V) \rangle.$$

In practical computations, it is IMPOSSIBLE to traverse all $f, g$. Some classical methods are to approximate HGR by requiring $f, g$ belonging to linear space or Reproducing Kernel Hilbert Spaces. In [4], the authors use an $m$-out-of-$n$ bootstrap to estimate the singular values of a stochastic matrix from a finite sample and solve it through a kernel approximation.

MI terms are DIFFICULT to estimate and optimize, researchers try to replace the MI by the lower/upper bounds [5] or some variational methods [6].

[1]. J. Liu, Z. Li, Y. Yao, F. Xu, X. Ma, M. Xu, and H. Tong. Fair Representation Learning: An Alternative to Mutual Information. SIGKDD, 1088–1097, 2022.

[2] E. Creager, D. Madras, J.-H. Jacobsen, M. Weis, K. Swersky, T. Pitassi, and R. Zemel. Flexibly fair representation learning by disentanglement. ICLR, 1436-1445, 2019.

[3] C. Louizos, K. Swersky, Y. Li, M. Welling, and R. Zemel. The variational fair autoencoder. arXiv preprint arXiv:1511.00830, 2015.

[4]. J. Mary, C. Calauzènes, and N. E. Karoui. Fairness-Aware Learning for Continuous Attributes and Treatments. ICML, 4382–4391, 2019.

[5]. J. Song, P. Kalluri, A. Grover, S. Zhao, and S. Ermon. Learning Controllable Fair Representations. AISTAT, 2164–2173, 2019.

[6] J. Song, and E. Stefano. Understanding the Limitations of Variational Mutual Information Estimators. ICLR, 2019.

4. In the experimental results (Table 1 and 2), it seems that the proposed results consistently have higher accuracy and lower fairness violation. However, the proposed result is not too different from other methods as most of them are in the Lagrangian form, i.e., CE loss plus fairness/ independence constrains. It is encouraged that the authors explain clearly why the proposed method could lead to a consistently better acc-fariness trade-off point than other methods.

**Answer:** Thanks for your comment. A consistently better result is from a better model + better model parameter selection + better optimization. As stated in the above, our DC regularization is a good choice for fair classification.

In our optimization problems, there is a crucial parameter that plays a significant role in achieving optimal performance. The selection and adjustment of this parameter greatly influence the optimization process and the quality of the resulting solution. To address this, we employ the Lagrangian dual method as an alternative approach to update the balanced parameter and network parameters.

In our experiments, we initially attempted to manually choose the balanced parameter. However, we found that the performance was slightly lower compared to using the Lagrangian dual method. This highlights the advantage of employing an adaptive approach.

## 2 REBUTTAL FOR REVIEWER eHoC

### 2.1 QUESTIONS MAIN ARGUMENTS

1. The novelty of the proposed fair classification method came from two parts:

(1) introduce an alternative independence approximation named distance covariance. For me, the first novelty is limited as this is an extension of previous work, e.g., mutual information (Kamishima et al., 2012), covariance (Zafar et al., 2017), or HGR coefficient ****(Mary et al., 2019) were token as the approximation metric. These works all added the empirical approximation to the original loss function as the regularization term.

As the authors mentioned in the paper, the covariance only captures linear dependency between sensitive attributes and the predictions. MI and HGR are also nonlinear independence approximation metrics. The superiority of selecting distance covariance over other nonlinear metrics are not clear in terms of computational efforts and math property.

In particular, Mutual Information is nonnegative measure, closely related to KL divergence measure, and can be well approximated by subsamples (Kamishima et al., 2012).

Thanks for your time to review our paper, we add their comparisons into the introduction section of the revised paper.

While alternatives such as MI, covariance, and HGR can potentially replace distance covariance (DC), it is important to note that DC offers several advantages that these three alternatives lack.

– Two random vectors (variables) are independent if and only if any of their HGR maximal correlation, MI, and distance covariance (DC) obtain a value of 0, but independence implies Covariance= 0 and the converse is not always true.
This implies that MI, HGR, **DC** can capture all relationship between the random vectors we used. Therefore, our DC can capture ALL relationships, but NOT only linear dependency.

– The definition of HGR of two random vectors is

$$HGR(U,V) = \sup_{f,g} \langle f(U), g(V) \rangle.$$

In practical computations, it is IMPOSSIBLE to traverse all $f, g$. Some classical methods are to approximate HGR by requiring $f, g$ belonging to linear space or Reproducing Kernel Hilbert Spaces. In [1], the authors solve it through a kernel approximation.

– MI terms are DIFFICULT to estimate and optimize, they replace the MI by the lower/upper bounds [2] or some variational methods [3]. In practice, this inevitably introduces biases and errors, subsequently reducing the utility of downstream classification tasks.

– For DC, empirical DC, as a statistic, can be directly computed from the samples, although population DC [3] is also challenge for computing since it requires knowledge of the analytical form of the distribution function and involves integration.
In addition, the empirical DC is a continuously differentiable and biconvex function about the predicted target matrix and the sensitive attribute matrix. In the fairness classification problem, the sensitive attribute matrix is known, so the empirical DC is a continuously differentiable convex function of the predicted target matrix, which is an elegant property.
Moreover, the paper [4] also demonstrates the advantages of DC by comparing with methods based on mutual information [5,6], demonstrating its advantages. We further utilized empirical distance covariance, not only reducing training time but also improving both accuracy and fairness.

[1]. J. Liu, Z. Li, Y. Yao, F. Xu, X. Ma, M. Xu, and H. Tong. Fair Representation Learning: An Alternative to Mutual Information. SIGKDD, 1088–1097, 2022.

[2]. J. Song, P. Kalluri, A. Grover, S. Zhao, and S. Ermon. Learning Controllable Fair Representations. AISTAT, 2164–2173, 2019.

[3] J. Song, and E. Stefano. Understanding the Limitations of Variational Mutual Information Estimators. ICLR, 2019.

[4]. J. Mary, C. Calauzènes, and N. E. Karoui. Fairness-Aware Learning for Continuous Attributes and Treatments. ICML, 4382–4391, 2019.

[5] E. Creager, D. Madras, J.-H. Jacobsen, M. Weis, K. Swersky, T. Pitassi, and R. Zemel. Flexibly fair representation learning by disentanglement. ICLR, 1436-1445, 2019.

[6] C. Louizos, K. Swersky, Y. Li, M. Welling, and R. Zemel. The variational fair autoencoder. arXiv preprint arXiv:1511.00830, 2015.

(2) leverage Lagrangian primal-dual alternative optimization to automatically select the weight coefficient. This part of contribution is debatable. While the Lagrangian approach is an efficient way of iteratively updating both training parameters and the weight coefficient, it also lost the advantage of controlling the trade-off if the decision-maker does have the domain knowledge.

**Answer**: Thanks for your comment.

Initially, we conducted experiments to optimize our problem using both manual parameter tuning and the Lagrangian dual method. Through these experiments, we observed that the performance of the Lagrangian dual method is better when both methods are from the same value of $\lambda$. This finding suggests that the Lagrangian dual method contributes to the stability of the optimization process and leads to improved results.

In the Primal Dual method, the decision-maker also has some control over the fairness-utility trade-off curve by choosing the initial guess of $\lambda$ or other hyperparameters.

2. The main theoretical contributions are the analysis of the properties of distance covariance and the convergence analysis of the empirical distance covariance. While the existence of bi-convexity renders lower effort in minimizing the distance covariance in the fair ML setting, the convergence analysis is not associated with the quality of the fair solution, i.e., how the convergence speed to a (Pareto) minimizer of the penalized training object is impacted by the sample size.

**Answer**: The convergence analysis of the empirical distance covariance holds significant importance for our fairness classification task. This is because:

- **Suitability**: In our task, it is crucial to assess the suitability of using empirical distance covariance as the regularized term. If there is a substantial difference between the empirical distance covariance and the population distance covariance, it may not be appropriate to rely on the empirical version. This is because the theorem we have only establishes that DC(predicted target, sensitive attribute) equals 0 if and only if the two variables are independent. If there is a significant discrepancy between the empirical and population distance covariances, using the empirical version could lead to biased or inaccurate results.

- **Consideration of Batch Computation**: In our deep learning computations, data is typically processed in batches rather than considering the entire dataset at once. As a result, the number of samples we can work with is limited and cannot tend to infinity. To address this constraint, our theorem (Theorem 4) provides valuable insights by specifying the minimum number of samples required to achieve a user-defined precision. This information allows us to determine the sample size needed to ensure reliable and accurate results in our fairness classification task.

The biconvexity of the empirical distance covariance in our objective function is a key characteristic that contributes to the elegance of our optimization problem.

Furthermore, analyzing the convergence speed represents a valuable future direction to enhance the computational efficiency and performance of our approach.

3. It is not clear in the paper whether the distance covariance has non-negative property. This is related to the sign of Lagrangian multiplier. It should not have any sign constraint for equality equality-constrained problem given in (2).

**Answer**: Both distance covariance and empirical distance covariance exactly have non-negative property. The definition of distance covariance (also see our paper) is

$$\mathcal{V}(\mathbf{Y}, \mathbf{Z}) = \int_{\mathbb{R}^{p+q}} |f_{\mathbf{Y},\mathbf{Z}}(t, s) - f_{\mathbf{Y}}(t)f_{\mathbf{Z}}(s)|^2 w(t, s) \, dt ds,$$

where the weight $w(t, s) = (c_p c_q |t|^{1+p} |s|^{1+q})^{-1}$ with $c_d = \frac{\pi^{(1+d)/2}}{\Gamma((1+d)/2)}$ and $\Gamma$ being the gamma function.

It is obvious the integated function is non-negative, so distance covariance is non-negative.

The non-negativity of empirical distance covariance can be guaranteed by Theorem 1 in [1].

[1] G. J. Székely, M. L. Rizzo, and N. K. Bakirov. Measuring and testing dependence by correlation of distances. The Annals of Statistics, 2769-2794, 2007.

## 2.2 QUESTIONS IMPRECISE PART

1. Section 4.2 P8: It is not clear how the trade-off curve is obtained if the Lagrangian multiplier is not under control. Fixing the Lagrangian multiplier or just randomly initialize the starting points?

**Answer**: The points of Figure 1 in Section 4.1 are related to the initial guesses of balanced parameter $\lambda$, since different inital guesses will give different results. In our experiments, the points we choose are in the interval $[1, 15]$ as the initial guesses.

2. Figure 1 Right P8: a trade-off curve should only contain non-dominated solutions. For example, the left most blue dot is dominated by the second dot and should not included in the numerical result.

**Answer**:Thank you for your comment, we delete the dominated one.

3. Section 4.2.2: Not sure if I understand correctly, the experiment targeting predicting gender is odd and not aligned with any realistic applications.

**Answer**: We actually follow the experimental setup from [1] and evaluate our proposed method by comparing with some baselines.

[1] Park, Sungho, et al. "Fair contrastive learning for facial attribute classification." Proceedings of the IEEE/CVF Conference on Computer Vision and Pattern Recognition. 2022.

4. It is mentioned in the Related Work section that benchmark methods like HGR, MI, etc. are computationally challenging. So it is natural that readers are expecting a comparison of computation effort.

**Answer**: Thanks for your comment, we provide the computational time about many algorithms, including HGR and FCRL (MI-based method), on the Adult dataset in Table 1.

The computation challenge in our paper refers to the impossibility of direct calculation of HGR and MI. Instead, estimations or lower/upper bounds need to be employed.

In the related works, we try to compare the three algorithms. We can calculate empirical DC directly, but only estimations of the others can be utilized for computatoin.

In reference [1], a kernel approximation is utilized for HGR and the lower and upper bounds of Mutual Information (MI) are utilized to deal with the computation challenge in [2, 3].

[1]. J. Mary, C. Calauzènes, and N. E. Karoui. Fairness-Aware Learning for Continuous Attributes and Treatments. ICML, 4382–4391, 2019.

| Adult | Time(second) |
|---|---|
| Vallina | 21.783 |
| Ours | 33.85 |
| HGR | 47.91 |
| Fair Mixup | 50.64 |
| FCRL | 684.03 |
| FairDisCo | 704.28 |
| Dist-fair | 479.78 |
| FSCL | 161.67 |
| FairProjection | 100.98 |
| FERMI | 642.43 |

Table 1: Computational time on the Adult dataset.

[2]. J. Song, P. Kalluri, A. Grover, S. Zhao, and S. Ermon. Learning Controllable Fair Representations. AISTAT, 2164–2173, 2019.

[3] Gupta, Umang, et al. "Controllable guarantees for fair outcomes via contrastive information estimation." Proceedings of the AAAI Conference on Artificial Intelligence. Vol. 35. No. 9. 2021.

# 3 REBUTTAL FOR REVIEWER uH68

## 3.1 WEAKNESS

W1. The paper is relatively weak in describing and analyzing the mathematical properties of distance covariance. Although the paper mentions that distance covariance can measure linear and nonlinear correlations between two random vectors (predicted values and sensitive attributes), are there other metrics that can also capture nonlinear relationships between variables?

Also, does distance covariance have any advantages over other measures? This is the starting point of why distance covariance is used as a constraint term, which is not elaborated in the article.

**Answer**: Thanks for your comment, we add the comparisons into the introduction part of the revised paper.

All of Distance covariance (DC), MI, and HGR can be used to capture linear and nonlinear correlations between two random vectors, but there are some advantages of DC in computation and optimization that the others do not have:

- For DC, Empirical DC, as a statistic, can be directly computed from the samples, although population DC [1] is also challenge for computing since it requires knowledge of the analytical form of the distribution function and involves integration.

  In addition, the empirical DC is a continuously differentiable and biconvex function about the predicted target matrix and the sensitive attribute matrix. In the fairness classification problem, the sensitive attribute matrix is known, so the empirical DC is a continuously differentiable convex function of the predicted target matrix, which is an elegant property.

  Moreover, the paper [1] also demonstrates the advantages of DC by comparing with methods based on mutual information [2,3], demonstrating its advantages. We further utilized empirical distance covariance, not only reducing training time but also improving both accuracy and fairness.

- For HGR and MI, we can only use the estimations or upper/lower bounds instead, which are used to approximate independence. In practice, this inevitably introduces biases and errors, subsequently reducing the utility of downstream classification tasks.

  The definition of HGR of two random vectors is

  $$HGR(U, V) = \sup_{f,g} \langle f(U), g(V) \rangle.$$

  In practical computations, it is IMPOSSIBLE to traverse all $f, g$. Some classical methods are to approximate HGR by requiring $f, g$ belonging to linear space or Reproducing Kernel Hilbert Spaces. In [4], the authors use an $m$-out-of-$n$ bootstrap to estimate the singular values of a stochastic matrix from a finite sample and solve it through a kernel approximation. .

  MI terms are DIFFICULT to estimate and optimize, they replace the MI by the lower/upper bounds [5] or some variational methods [6].

[1]. J. Liu, Z. Li, Y. Yao, F. Xu, X. Ma, M. Xu, and H. Tong. Fair Representation Learning: An Alternative to Mutual Information. SIGKDD, 1088–1097, 2022.

[2] E. Creager, D. Madras, J.-H. Jacobsen, M. Weis, K. Swersky, T. Pitassi, and R. Zemel. Flexibly fair representation learning by disentanglement. ICLR, 1436-1445, 2019.

[3] C. Louizos, K. Swersky, Y. Li, M. Welling, and R. Zemel. The variational fair autoencoder. arXiv preprint arXiv:1511.00830, 2015.

[4]. J. Mary, C. Calauzènes, and N. E. Karoui. Fairness-Aware Learning for Continuous Attributes and Treatments. ICML, 4382–4391, 2019.

[5]. J. Song, P. Kalluri, A. Grover, S. Zhao, and S. Ermon. Learning Controllable Fair Representations. AISTAT, 2164–2173, 2019.

[6] J. Song, and E. Stefano. Understanding the Limitations of Variational Mutual Information Estimators. ICLR, 2019.

**W2.** The explanation of the connection between DP and EO in Section 3.3 is not very clear. For example, "the equation (7) suggests that the objective goes beyond achieving independence between the feature representation $\phi_\theta(X)$ and the sensitive attribute Z." Is $\phi_\theta(X)$ a feature representation or a prediction?

**Answer**: Sorry, we made a mistake of the statement, $\phi_\theta(X)$ is a prediction. Thanks for your reminder.

In Section 3.3, we try to think about the connections between DP and EO and find: Model $\max_\theta P(\phi_\theta(X) = Y), \text{s.t.} \ \phi_\theta(X) \perp Z$ suggests that the objective goes beyond achieving independence between the predictions $\phi_\theta(X)$ and the sensitive attribute $Z$.

Intuitively, samples sharing the same sensitive attribute, regardless of their target classes, have a tendency to cluster together due to shared characteristics or patterns within those groups. Conversely, the fitting term related to the target attribute places greater emphasis on accurately classifying the majority group. This is because capturing the patterns and characteristics of the majority group is often more crucial for optimizing the model's overall performance.

Suppose $Y = y$ is a majority class in the sensitive class $Z = Z_i$, but not a majority class in $Z = Z_j$. The worst case is $P(\hat{Y} = y|Y = y, Z = Z_i) = 1$ and $P(\hat{Y} = y|Y = y, Z = Z_j) = 0$ since $Y = y$ is not a majority class in $Z = A_j$, which implies strong dependence between $Y$ and $Z$. The introduction of independence seeks to break this dependence, leading to an increase in $P(\hat{Y} = y|Y = y, Z = Z_j)$, and resulting in a smaller EO value.

We also provide numerical illustrations on the connection between DP and EO in the Appendix B.

In Figure 2, we showcase the model's performance on the test set of the CelebA dataset across various values of the parameter $\lambda$. As the value of $\lambda$ increases, both the accuracy (Acc) and the differential privacy gap ($\Delta$DP) decrease. Conversely, the equal opportunity gap ($\Delta$EO) exhibits a $V$-shaped or increasing trend. Based on these observations, we can numerically select an optimal parameter value such that both DP and EO are smaller while maintaining a high level of predictive accuracy.

We also provide detailed insights into the UTKFace dataset. Despite manually introducing imbalances in the dataset, we observed that both DP and EO metrics decrease simultaneously as the number of epochs increases, while accuracy improves.

In our setting for each sensitive attribute class, the ratio of the majority class to the minority class in relation to the target classes is the same. This observation can possibly be attributed to the fact that we maintained a consistent imbalance factor across all sensitive attribute classes.

**W3.** In Experiment 4.1, the trend between accuracy and EO demonstrated by the proposed method is quite different from the comparison method. Is there any analysis to explain this?

**Answer**: In the paper, the right-hand side (now the second subfigure in the revised version) of Figure 1 displays points representing (accuracy, EO)-pairs with different initial guesses of the balanced parameter $\lambda \in [1, 15]$. As mentioned in the second reviewer, it may be better to remove the leftmost blue dot.

We appreciate the comment and will make the necessary adjustments in the revised paper. Thank you for your comment.

**W4.** Does this method work equally well in scenarios where the sensitive attribute is a continuous variable?

**Answer**: Indeed, our empirical distance covariance is applicable to both discrete and continuous variables.

To the best of our knowledge, all datasets containing continuous sensitive attributes are currently used for regression tasks, such as the Communities and Crime dataset mentioned in [1]. As our method in this paper focuses on achieving fairness in

classification, we have identified fair regression as a potential avenue for future research.

[1]. J. Mary, C. Calauzènes, and N. E. Karoui. Fairness-Aware Learning for Continuous Attributes and Treatments. ICML, 4382–4391, 2019.

## 3.2  QUESTIONS

Explain the proposd connection between DP and EO.

Does this method work equally well in scenarios where the sensitive attribute is a continuous variable?

# 4 REBUTTAL FOR REVIEWER zY1Z

## 4.1 WEAKNESS

W1. Section 3.3 claims that optimizing for Eq. (6) leads to an optimal case where both DP and EO are satisfied. This contradicts to the fact that (also mentioned in Section 3.3) "It is only possible to achieve both DP and EO when the sensitive attributes Z are independent of the labels Y." Therefore, I do not think it justifies the claim that the proposed approach can achieve both DP and EO at the same time.

W3. More tabular data experiments should be conducted to confirm the finding. E.g. on datasets provided by [1].
[1] Ding, Frances, Moritz Hardt, John Miller, and Ludwig Schmidt. "Retiring adult: New datasets for fair machine learning." Advances in neural information processing systems 34 (2021): 6478-6490.

## 4.2 QUESTIONS

Can you discuss more about why the proposed approach can achieve good EO results? This finding is both interesting and suspicious to me. Given that "It is only possible to achieve both DP and EO when the sensitive attributes Z are independent of the labels Y," it is not likely that good EO should be achieved when optimizing with the constraint $\xi(X) \perp Z$.

**Answer**: Our statement: EO will be improved when optimizing the model $\max_\theta P(\phi_\theta(X) = Y), \text{s.t. } \phi_\theta(X) \perp Z$.

In Section 3.3, we try to think about the connections between DP and EO and find:
Model $\max_\theta P(\phi_\theta(X) = Y), \text{s.t. } \phi_\theta(X) \perp Z$ suggests that the objective goes beyond achieving independence between the predictions $\phi_\theta(X)$ and the sensitive attribute $Z$.

Intuitively, samples sharing the same sensitive attribute, regardless of their target classes, have a tendency to cluster together due to shared characteristics or patterns within those groups. Conversely, the fitting term related to the target attribute places greater emphasis on accurately classifying the majority group. This is because capturing the patterns and characteristics of the majority group is often more crucial for optimizing the model's overall performance.

Suppose $Y = y$ is a majority class in the sensitive class $Z = Z_i$, but not a majority class in $Z = Z_j$. The worst case is $P(\hat{Y} = y | Y = y, Z = Z_i) = 1$ and $P(\hat{Y} = y | Y = y, Z = Z_j) = 0$ since $Y = y$ is not a majority class in $Z = A_j$, which implies strong dependence between $Y$ and $Z$. The introduction of independence seeks to break this dependence, leading to an increase in $P(\hat{Y} = y | Y = y, Z = Z_j)$, and resulting in a smaller EO value.

We also provide numerical illustrations on the connection between DP and EO in the Appendix B.

In Figure 2, we showcase the model's performance on the test set of the CelebA dataset across various values of the parameter $\lambda$. As the value of $\lambda$ increases, both the accuracy (Acc) and the differential privacy gap ($\Delta$DP) decrease. Conversely, the equal opportunity gap ($\Delta$EO) exhibits a $V$-shaped or increasing trend. Based on these observations, we can numerically select an optimal parameter value such that both DP and EO are smaller while maintaining a high level of predictive accuracy.

We also provide detailed insights into the UTKFace dataset. Despite manually introducing imbalances in the dataset, we observed that both DP and EO metrics decrease simultaneously as the number of epochs increases, while accuracy improves.

In our setting for each sensitive attribute class, the ratio of the majority class to the minority class in relation to the target classes is the same. This observation can possibly be attributed to the fact that we maintained a consistent imbalance factor across all sensitive attribute classes.

E.g. can you also calculate the empirical distance covariance between Y and Z for your datasets?

**Answer**: The empirical distance covariance is based on Definition 3.1 in our paper. Let $Y = [Y_1, \cdots, Y_n]$ and $Z = [Z_1, \cdots, Z_n]$ be the sample matrices, where $n$ is the sample number. Then we can calculate $a_{kl} = \|Y_k - Y_l\|_2$, $\bar{a}_{k.} = \frac{1}{n} \sum_{l=1}^{n} a_{kl}$, $\bar{a}_{.l} = \frac{1}{n} \sum_{k=1}^{n} a_{kl}$, $\bar{a}_{..} = \frac{1}{n^2} \sum_{l,k=1}^{n} a_{kl}$ and $b_{kl} = \|Z_k - Z_l\|_2$, $\bar{b}_{k.} = \frac{1}{n} \sum_{l=1}^{n} b_{kl}$, $\bar{b}_{.l} = \frac{1}{n} \sum_{k=1}^{n} b_{kl}$, $\bar{b}_{..} = \frac{1}{n^2} \sum_{l,k=1}^{n} b_{kl}$. The empirical distance covariance $\mathcal{V}_n(Y, Z)$ is

$$\mathcal{V}_n(Y, Z) = \frac{1}{n^2} \sum_{k,l=1}^{n} A_{kl} B_{kl}, \tag{1}$$

where $A_{kl} = a_{kl} - \bar{a}_{k.} - \bar{a}_{.l} + \bar{a}_{..}$, $B_{kl} = b_{kl} - \bar{b}_{k.} - \bar{b}_{.l} + \bar{b}_{..}$.

The non-negativity of empirical distance covariance can be guaranteed by Theorem 1 in [1].

[1] G. J. Székely, M. L. Rizzo, and N. K. Bakirov. Measuring and testing dependence by correlation of distances. The Annals of Statistics, 2769-2794, 2007.

I would also suggest the authors to consider adding more baselines and experiment on more datasets to further confirm this finding (as discussed in my weaknesses).

**Answer**: Thanks for your suggestion, we add numerical experiments about new dataset in our revised paper.

In Table 2, why are the results for EO and DP separated (and Acc for $\alpha$=2 are different for Col 2 and Col 8)? Table 1 shows those results together.

**Answer**: For the UTKFace dataset, we conducted experiments for 90 epochs using all the methods listed in Table 2. The results reported in Table 2 is the optimal performance achieved during the last 10 epochs. That is the reason why there is a slight difference. We will modify this part (see our revised paper).

In comparing the submitted and revised papers, it is important to note that there are a few differences concerning the FSCL and Dist-Fair algorithms. This is primarily due to the unavailability of the results for these algorithms in the revised paper, as we were unable to retrieve the previous results and rerun the code.

W2. Most of the baselines optimize for DP (only FSCL optimizes for EO and is primarily tailored for image datasets). More baseline methods specifically optimizing for EO should be included.

**Answer**: Thanks for your comment. Except for FSCL, both FairMixup and HGR in baselines actually optimize for EO for all experiments but two Accuracy-DP trade-off curves on tabular datasets.