# OpenReview forum: "Fair Attribute Classification via Distance Covariance"
_ICLR.cc/2024/Conference — Submitted to ICLR 2024_

### Official Review · Reviewer_zY1Z · 2023-10-16

**Soundness:** 2 fair
**Presentation:** 3 good
**Contribution:** 2 fair
**Rating:** 5
**Confidence:** 3

**Summary:**

In order to improve independence between a model's predictions and sensitive attributes, this paper utilizes empirical distance as a constraint.  This constrained problem is then optimized using the Lagrangian dual method to find a better trade-off between accuracy and
fairness.

**Strengths:**

1. Results show that the proposed approach can achieve good DP and EO at the same time while maintaining a good accuracy. Which are not commonly seen results.

2. Theorem 4 provides the convergence of using empirical distance covariance to estimate population distance covariance.

3. The proposed solution is technically sound.

4. The presentation is clear.

**Weaknesses:**

1. Section 3.3 claims that optimizing for Eq. (6) leads to an optimal case where both DP and EO are satisfied. This contradicts to the fact that (also mentioned in Section 3.3) "It is only possible to achieve both DP and EO when the sensitive attributes Z are independent of the labels Y." Therefore, I do not think it justifies the claim that the proposed approach can achieve both DP and EO at the same time.

2. Most of the baselines optimize for DP (only FSCL optimizes for EO and is primarily tailored for image datasets). More baseline methods specifically optimizing for EO should be included.

3. More tabular data experiments should be conducted to confirm the finding. E.g. on datasets provided by [1].

[1] Ding, Frances, Moritz Hardt, John Miller, and Ludwig Schmidt. "Retiring adult: New datasets for fair machine learning." Advances in neural information processing systems 34 (2021): 6478-6490.

**Questions:**

1. Can you discuss more about why the proposed approach can achieve good EO results? This finding is both interesting and suspicious to me. Given that "It is only possible to achieve both DP and EO when the sensitive attributes Z are independent of the labels Y," it is not likely that good EO should be achieved when optimizing with the constraint  ϕ(X) ⊥ Z. E.g. can you also calculate the empirical distance covariance between Y and Z for your datasets? I would also suggest the authors to consider adding more baselines and experiment on more datasets to further confirm this finding (as discussed in my weaknesses).

2. In Table 2, why are the results for EO and DP separated (and Acc for α=2 are different for Col 2 and Col 8)? Table 1 shows those results together.

---

> ### Author Response · Authors · 2023-11-16
> **Explanation of good EO and nonnegativity of dc**
>
> **1-1. Can you discuss more about why the proposed approach can achieve good EO results? This finding is both interesting and suspicious to me. Given that "It is only possible to achieve both DP and EO when the sensitive attributes Z are independent of the labels Y," it is not likely that good EO should be achieved when optimizing with the constraint $\xi(X) \perp Z$.**
>
> **Answer**: Our statement: EO will be improved when optimizing the model $\underset{\theta}{\mathrm{max}} \: P(\phi_\theta(X)=Y),
> 	\mathrm{s.t.}\ \phi_\theta(X) \perp Z$.
>
> In Section 3.3, we try to think about the connections between DP and EO and find:
>
> Model $\underset{\theta}{\mathrm{max}} \: P(\phi_\theta(X)=Y),
> 	\mathrm{s.t.}\ \phi_\theta(X) \perp Z$ suggests that **the objective goes beyond achieving independence between the predictions $\phi_\theta(X)$ and the sensitive attribute $Z$.**
>
> Intuitively, samples sharing the same sensitive attribute, regardless of their target classes, have a tendency to cluster together due to shared characteristics or patterns within those groups. Conversely, the fitting term related to the target attribute places greater emphasis on accurately classifying the majority group. This is because capturing the patterns and characteristics of the majority group is often more crucial for optimizing the model's overall performance.
>
> Suppose $Y=y$ is a majority class in the sensitive class $Z=Z_i$, but not a majority class in $Z=Z_j$. The worst case is $P(\hat Y=y|Y=y,Z=Z_i)=1$ and $P(\hat Y=y|Y=y,Z=Z_j)=0$ since $Y=y$ is not a majority class in $Z=A_j$, which implies strong dependence between $Y$ and $Z$. **The introduction of independence seeks to break this dependence**, leading to an increase in $P(\hat Y=y|Y=y,Z=Z_j)$, and resulting in a smaller EO value.
>
> We also provide numerical illustrations on the connection between DP and EO in the Appendix B.
>
> In Figure 2, we showcase the model's performance on the test set of the CelebA dataset across various values of the parameter $\lambda$. As the value of $\lambda$ increases, both the accuracy (Acc) and the differential privacy gap ($\Delta$DP) decrease. Conversely, the equal opportunity gap ($\Delta$EO) exhibits a $V$-shaped or increasing trend. Based on these observations, we can numerically select an optimal parameter value such that both DP and EO are smaller while maintaining a high level of predictive accuracy.
>
> We also provide detailed insights into the UTKFace dataset. Despite manually introducing imbalances in the dataset, we observed that both DP and EO metrics decrease simultaneously as the number of epochs increases, while accuracy improves.
>
> In our setting for each sensitive attribute class, the ratio of the majority class to the minority class in relation to the target classes is the same. This observation can possibly be attributed to the fact that we maintained a consistent imbalance factor across all sensitive attribute classes.
>
> **1-2. E.g. can you also calculate the empirical distance covariance between Y and Z for your datasets?**
>
> **Answer**: The empirical distance covariance is based on Definition 3.1 in our paper.
>
> Let $Y=[Y_1,\cdots,Y_n]$ and $Z=[Z_1,\cdots,Z_n]$ be the sample matrices, where $n$ is the sample number. Then we can calculate $a_{kl}  = \Vert Y_k-Y_l \Vert_2$, $\bar a_{k\cdot}  = \frac{1}{n}\sum_{l=1}^{n}a_{kl}$,
>  $\bar a_{\cdot l}  = \frac{1}{n}\sum_{k=1}^{n}a_{kl}$,
> $\bar a_{\cdot \cdot}  = \frac{1}{n^2}\sum_{l,k=1}^{n}a_{kl}$
>  and
> $b_{kl} = \Vert Z_k-Z_l \Vert_2$,
> $\bar b_{k\cdot}  = \frac{1}{n}\sum_{l=1}^{n}b_{kl}$,
> $\bar b_{\cdot l}  = \frac{1}{n}\sum_{k=1}^{n}b_{kl}$,
> $\bar b_{\cdot \cdot}  = \frac{1}{n^2}\sum_{l,k=1}^{n}b_{kl}$.
>
> The empirical distance covariance $\mathcal{V}_n(Y,Z)$ is $\mathcal{V}_n(Y,Z) = \frac1{n^2}\langle A,B\rangle$
>
> where
> $A_{kl}  = a_{kl} - \bar a_{k\cdot} - \bar a_{\cdot l} + \bar a_{\cdot\cdot}$, $B_{kl} = b_{kl} - \bar b_{k\cdot} - \bar b_{\cdot l} + \bar b_{\cdot\cdot}$.
>
>  The non-negativity of empirical distance covariance can be guaranteed by Theorem 1 in [1].
>
> [1] G. J. Szekely, M. L. Rizzo, and N. K. Bakirov. Measuring and testing dependence by correlation of distances. The Annals of Statistics, 2769-2794, 2007.
>
> **1-3. I would also suggest the authors to consider adding more baselines and experiment on more datasets to further confirm this finding (as discussed in my weaknesses).**
>
> **Answer**: Thanks for your suggestion, we add numerical experiments about new dataset in our revised paper.

---

> ### Author Response · Authors · 2023-11-16
> **Response for Fair Attribute Classification via Distance Covariance**
>
> **2. In Table 2, why are the results for EO and DP separated (and Acc for $\alpha$=2 are different for Col 2 and Col 8)? Table 1 shows those results together.**
>
> **Answer**: For the UTKFace dataset, we conducted experiments for 90 epochs using all the methods listed in Table 2. The results reported in Table 2 is the optimal performance achieved during the last 10 epochs. That is the reason why there is a slight difference. We will modify this part (see our revised paper).
>
> In comparing the submitted and revised papers, it is important to note that there are a few differences concerning the FSCL and Dist-Fair algorithms. This is primarily due to the unavailability of the results for these algorithms in the revised paper, as we were unable to retrieve the previous results and rerun the code.
>
> **W2. Most of the baselines optimize for DP (only FSCL optimizes for EO and is primarily tailored for image datasets). More baseline methods specifically optimizing for EO should be included.**
>
> **Answer**: Thanks for your comment. Except for FSCL, both FairMixup and HGR in baselines actually optimize for EO for all experiments but two Accuracy-DP trade-off curves on tabular datasets.
>
> *Hope our reply solves your concern. We are appreciated you can change you decision and give us more score.*

---

### Official Review · Reviewer_uH68 · 2023-10-18

**Soundness:** 3 good
**Presentation:** 3 good
**Contribution:** 2 fair
**Rating:** 5
**Confidence:** 4

**Summary:**

In this paper, the (empirical) distance covariance is introduced as a fairness constraint term for achieving group fairness in fair classification tasks. During optimization, the Lagrange dual method is used, and automatic hyperparameter selection is implemented. The authors also provide an estimation of the difference between the empirical distance covariance and the population distance covariance. Experiments are conducted on the tabular dataset and the image dataset, respectively, to show the performance of the proposed method.

**Strengths:**

S1. The introduction of distance covariance as a fairness constraint in the fair classification.
S2. The paper provides some theoretical support for the computation of empirical distance covariance.
S3. The authors conducted experiments on both tabular and image datasets, as well as experimental comparisons in scenarios with multiple sensitive attributes and unbalanced distribution of data across different subgroups.

**Weaknesses:**

W1. The paper is relatively weak in describing and analyzing the mathematical properties of distance covariance. Although the paper mentions that distance covariance can measure linear and nonlinear correlations between two random vectors (predicted values and sensitive attributes), are there other metrics that can also capture nonlinear relationships between variables? Also, does distance covariance have any advantages over other measures? This is the starting point of why distance covariance is used as a constraint term, which is not elaborated in the article.

W2. The explanation of the connection between DP and EO in Section 3.3 is not very clear. For example, "the equation (7) suggests that the objective goes beyond achieving independence between the feature representation φ_θ(X) and the sensitive attribute Z." Is φ_θ(X) a feature representation or a prediction?

W3. In Experiment 4.1, the trend between accuracy and ∆EO demonstrated by the proposed method is quite different from the comparison method. Is there any analysis to explain this?

W4. Does this method work equally well in scenarios where the sensitive attribute is a continuous variable?

**Questions:**

1) Explain the proposd connection between DP and EO.
2) Does this method work equally well in scenarios where the sensitive attribute is a continuous variable?

---

> ### Author Response · Authors · 2023-11-16
> **The mathematical properties of distance covariance**
>
> **W1. The paper is relatively weak in describing and analyzing the mathematical properties of distance covariance. Although the paper mentions that distance covariance can measure linear and nonlinear correlations between two random vectors (predicted values and sensitive attributes), are there other metrics that can also capture nonlinear relationships between variables? Also, does distance covariance have any advantages over other measures? This is the starting point of why distance covariance is used as a constraint term, which is not elaborated in the article.**
>
> **Answer**: Thanks for your comment, we add the comparisons into the introduction part of the revised paper.
>
> All of Distance covariance (DC), MI,  and HGR can be used to capture linear and nonlinear correlations between two random vectors, but there are some advantages of DC in computation and optimization that the others do not have:
>
> - For DC, Empirical DC, as a statistic, can be **directly** computed from the samples, although population DC [1] is also challenge for computing since it requires knowledge of the analytical form of the distribution function and involves integration.
>
> - In addition, the empirical DC is a **continuously differentiable and biconvex** function about the predicted target matrix and the sensitive attribute matrix. In the fairness classification problem, the sensitive attribute matrix is known, so the empirical DC is a continuously differentiable convex function of the predicted target matrix, which is an elegant property.
>
> - Moreover, the paper [1] also demonstrates the advantages of DC by comparing with methods based on mutual information [2,3], demonstrating its advantages. We further utilized empirical distance covariance, not only reducing training time but also improving both accuracy and fairness.
>
> - For HGR and MI, we can only use the **estimations or upper/lower bounds** instead, which are used to approximate independence. In practice, this inevitably introduces biases and errors, subsequently reducing the utility of downstream classification tasks.
>
> - The definition of HGR of two random vectors is
> $$HGR(U,V)=\sup_{f,g}\ \langle f(U),g(V)\rangle.$$
> In practical computations, it is **IMPOSSIBLE** to traverse all $f,g$. Some classical methods are to approximate HGR by requiring $f,g$ belonging to linear space or Reproducing Kernel Hilbert Spaces. In [4], the authors use an $m$-out-of-$n$ bootstrap to estimate the singular values of a stochastic matrix from a finite sample and solve it through a kernel approximation.
> .
>
> - MI terms are **DIFFICULT** to estimate and optimize, they replace the MI by the lower/upper bounds [5] or some variational methods [6].
>
>
> [1]. J. Liu, Z. Li, Y. Yao, F. Xu, X. Ma, M. Xu, and H. Tong. Fair Representation
> Learning: An Alternative to Mutual Information. SIGKDD, 1088–1097, 2022.
>
> [2] E. Creager, D. Madras, J.-H. Jacobsen, M. Weis, K. Swersky, T. Pitassi, and R. Zemel. Flexibly fair representation learning by disentanglement. ICLR, 1436-1445, 2019.
>
> [3] C. Louizos,  K. Swersky, Y. Li, M. Welling, and R. Zemel. The variational fair autoencoder. arXiv preprint arXiv:1511.00830, 2015.
>
> [4]. J. Mary, C. Calauzenes, and N. E. Karoui. Fairness-Aware Learning for
> Continuous Attributes and Treatments. ICML, 4382–4391, 2019.
>
> [5]. J. Song, P. Kalluri, A. Grover, S. Zhao, and S. Ermon. Learning
> Controllable Fair Representations. AISTAT, 2164–2173, 2019.
>
> [6] J. Song, and E. Stefano. Understanding the Limitations of Variational Mutual Information Estimators. ICLR, 2019.

---

> ### Author Response · Authors · 2023-11-16
> **The explanation of the connection between DP and EO**
>
> **W2. The explanation of the connection between DP and EO in Section 3.3 is not very clear. For example, "the equation (7) suggests that the objective goes beyond achieving independence between the feature representation $\phi_\theta(X)$ and the sensitive attribute Z." Is $\phi_\theta(X)$ a feature representation or a prediction?**
>
> **Answer**: Sorry, we made a mistake of the statement, $\phi_\theta(X)$ is a prediction. Thanks for your reminder.
>
> In Section 3.3, we try to think about the connections between DP and EO and find:
>
> Model $\underset{\theta}{\mathrm{max}} \: P(\phi_\theta(X)=Y),
> 	\mathrm{s.t.}\ \phi_\theta(X) \perp Z$ suggests that the **objective goes beyond achieving independence between the predictions $\phi_\theta(X)$ and the sensitive attribute $Z$.**
>
> Intuitively, samples sharing the same sensitive attribute, regardless of their target classes, have a tendency to cluster together due to shared characteristics or patterns within those groups. Conversely, the fitting term related to the target attribute places greater emphasis on accurately classifying the majority group. This is because capturing the patterns and characteristics of the majority group is often more crucial for optimizing the model's overall performance.
>
> Suppose $Y=y$ is a majority class in the sensitive class $Z=Z_i$, but not a majority class in $Z=Z_j$. The worst case is $P(\hat Y=y|Y=y,Z=Z_i)=1$ and $P(\hat Y=y|Y=y,Z=Z_j)=0$ since $Y=y$ is not a majority class in $Z=A_j$, which implies strong dependence between $Y$ and $Z$. The introduction of independence seeks to break this dependence, leading to an increase in $P(\hat Y=y|Y=y,Z=Z_j)$, and resulting in a smaller EO value.
>
> We also provide numerical illustrations on the connection between DP and EO in the Appendix B.
>
> In Figure 2, we showcase the model's performance on the test set of the CelebA dataset across various values of the parameter $\lambda$. As the value of $\lambda$ increases, both the accuracy (Acc) and the differential privacy gap ($\Delta$DP) decrease. Conversely, the equal opportunity gap ($\Delta$EO) exhibits a $V$-shaped or increasing trend. Based on these observations, we can numerically select an optimal parameter value such that both DP and EO are smaller while maintaining a high level of predictive accuracy.
>
> We also provide detailed insights into the UTKFace dataset. Despite manually introducing imbalances in the dataset, we observed that both DP and EO metrics decrease simultaneously as the number of epochs increases, while accuracy improves.
>
> In our setting for each sensitive attribute class, the ratio of the majority class to the minority class in relation to the target classes is the same. This observation can possibly be attributed to the fact that we maintained a consistent imbalance factor across all sensitive attribute classes.
>
> **W3. In Experiment 4.1, the trend between accuracy and EO demonstrated by the proposed method is quite different from the comparison method. Is there any analysis to explain this?**
>
> **Answer**: In the paper, the right-hand side (now the second subfigure in the revised version) of Figure 1 displays points representing (accuracy, EO)-pairs with different initial guesses of the balanced parameter $\lambda\in[1,15]$. As mentioned in the second reviewer, it may be better to remove the leftmost blue dot.
>
> We appreciate the comment and will make the necessary adjustments in the revised paper. Thank you for your comment.
>
> **W4. Does this method work equally well in scenarios where the sensitive attribute is a continuous variable?**
>
> **Answer**: Indeed, our empirical distance covariance is applicable to both discrete and continuous variables.
>
> To the best of our knowledge, all datasets containing continuous sensitive attributes are currently used for regression tasks, such as the Communities and Crime dataset mentioned in [1]. As our method in this paper focuses on achieving fairness in classification, we have identified fair regression as a potential avenue for future research.
>
> [1]. J. Mary, C. Calauzenes, and N. E. Karoui. Fairness-Aware Learning for
> Continuous Attributes and Treatments. ICML, 4382–4391, 2019.
>
> *Hope our reply can solve your concern. We are appreciated if you can change your decision and give us more score.*

---

> > ### Comment · Reviewer_uH68 · 2023-11-22
> > **Feedback to the authors**
> >
> > Dear authors,
> >
> > Thanks for you response. My questions are partially addressed. However, more mathematical or theoretical analysis is still appreciated on the properties of (empirical) distance covariance. This could better demonstrate the advantages and disadvantages when using (empirical) distance covariance.

---

> > > ### Author Response · Authors · 2023-11-22
> > > **Response to mathematical or theoretical analysis of the properties of (empirical) distance covariance**
> > >
> > > Dear reviewer,
> > >
> > > Thank you for taking the time to review our paper. We sincerely appreciate your valuable comments, which will undoubtedly contribute to improving the quality of our work.
> > >
> > > The concept of (empirical) distance covariance was first introduced in 2007 [1]. Since then, extensive research has been conducted on the mathematical and theoretical properties of covariance distance within the field of statistics.
> > >
> > > These properties, particularly relevant to fairness, encompass **independence** (as presented in Lemma 4 of our revised paper or Theorem 3 in [1]), **almost sure convergence** (as demonstrated in Lemma 2 of our paper or Theorem 2 in [1]) and **non-negativity**.
> > >
> > > The non-negativity of distance covariance can be derived from its definition, as the integrated function is inherently non-negative. Moreover, the non-negativity of empirical distance covariance is proven in Theorem 1 of [1], which we also emphasize in subsection 3.1 of our paper.
> > >
> > > Furthermore, we present the convergence result in terms of sample size in Theorem 3.
> > >
> > > In this paper, the use of empirical distance covariance offers several advantages. Firstly, it provides a means to **characterize independence**, which is a crucial aspect in statistical analysis. Secondly, empirical distance covariance offers **computational and optimization advantages**, making it a practical tool for various applications.
> > >
> > > However, it is important to note that empirical distance covariance is not suitable for addressing sufficiency in fairness problems, such as conditional use accuracy equality.
> > >
> > > [1] G. J. Szekely, M. L. Rizzo, and N. K. Bakirov. Measuring and testing dependence by correlation of distances. The Annals of Statistics, 2769-2794, 2007.

---

### Official Review · Reviewer_eHoC · 2023-10-31

**Soundness:** 3 good
**Presentation:** 3 good
**Contribution:** 2 fair
**Rating:** 5
**Confidence:** 3

**Summary:**

This paper leveraged empirical distance covariance as an approximation of statistic independence between two random variables. They incorporated it as an equality constraint in the classification tasks and solved it via Lagrangian dual approach without the need of manually specifying the weight. The authors conducted numerical experiments using different datasets and machine learning tasks, showing competitive performance over existing methods. The authors consolidated the detailed properties of the selected estimation metric distance covariance and provided a math proof of the convergence of empirical distance covariance to the true distance covariance in terms of sample size.

**Strengths:**

The paper is well-organized, well-written, and easy to follow. The numerical experiments demonstrated the superiority of the proposed empirical distance covariance over other approximation metrics in the literature.

**Weaknesses:**

I am giving a weakly reject because (1) the novelty is limited (see the main argument 1 for details). (2) the theoretical contribution is not targeted at the conference audiences. Theorem 4 delivers limited insight into the convergence speed of the proposed approach to the optimizer in terms of sample size (see the main argument 2). (3) The numerical experiment results demonstrated the superior performance of the new metric that is able to capture independence in a more accurate way. However, the computation cost was not compared.

**Questions:**

**Main arguments**

1. The novelty of the proposed fair classification method came from two parts: (1) introduce an alternative independence approximation named distance covariance. For me, the first novelty is limited as this is an extension of previous work, e.g., mutual information (Kamishima et al., 2012), covariance (Zafar et al., 2017), or HGR coefficient ****(Mary et al., 2019) were token as the approximation metric. These works all added the empirical approximation to the original loss function as the regularization term. As the authors mentioned in the paper, the covariance only captures linear dependency between sensitive attributes and the predictions. MI and HGR are also nonlinear independence approximation metrics. The superiority of selecting distance covariance over other nonlinear metrics are not clear in terms of computational efforts and math property. In particular, Mutual Information is nonnegative measure, closely related to KL divergence measure, and can be well approximated by subsamples (Kamishima et al., 2012). (2) leverage Lagrangian primal-dual alternative optimization to automatically select the weight coefficient. This part of contribution is debatable. While the Lagrangian approach is an efficient way of iteratively updating both training parameters and the weight coefficient, it also lost the advantage of controlling the trade-off if the decision-maker does have the domain knowledge.
2. The main theoretical contributions are the analysis of the properties of distance covariance and the convergence analysis of the empirical distance covariance. While the existence of bi-convexity renders lower effort in minimizing the distance covariance in the fair ML setting, the convergence analysis is not associated with the quality of the fair solution, i.e., how the convergence speed to a (Pareto) minimizer of the penalized training object is impacted by the sample size.
3. It is not clear in the paper whether the distance covariance has non-negative property. This is related to the sign of Lagrangian multiplier $\lambda$. It should not have any sign constraint for equality equality-constrained problem given in (2).

**The paper has some imprecise parts, here are a few:**

1. Section 4.2 P8: It is not clear how the trade-off curve is obtained if the Lagrangian multiplier is not under control. Fixing the Lagrangian multiplier or just randomly initialize the starting points?
2. Figure 1 Right P8: a trade-off curve should only contain non-dominated solutions. For example, the left most blue dot is dominated by the second dot and should not included in the numerical result.
3. Section 4.2.2: Not sure if I understand correctly, the experiment targeting predicting gender is odd and not aligned with any realistic applications.
4. It is mentioned in the Related Work section that benchmark methods like HGR, MI, etc. are computationally challenging. So it is natural that readers are expecting a comparison of computation effort.

**Minor Issues, no impact on the evaluation score:**

1. P7 Section 4 the 1st Paragraph: “The criteria used to assess the performance of fairness are…” reference format Park et al. (2022) needs to be corrected.
2. Section 4.2 P8: “Furthermore, there may be a mistake in
the implementation of the function *dis* within the provided code that computes the corresponding
probability density function.” If this is the case, I would remove FairDisCo from the comparison in Section 4.1 as well.

---

### Official Review · Reviewer_8JtY · 2023-10-31

**Soundness:** 3 good
**Presentation:** 3 good
**Contribution:** 3 good
**Rating:** 5
**Confidence:** 4

**Summary:**

This paper promotes the independence of the sensitive attribute and the predicted label to enhance fairness in classification by using (sample) distance covariance as a penalty term. The authors not only provide theoretical analysis on the convergence and sample complexity bounds for the estimation of distance covariance and mini-batch computation, but also numerical results on UCI tabular and image datasets.

**Strengths:**

1.	Distance covariance seems to be a very interesting metric for the dependency between two random variables and an active research area. So this paper which connect the distance covariance with fairness notion is very timely.
2.	This paper provides theoretical background, consistency and sample complexity bounds for the distance covariance
3.	Section 3.3 that connect DP and EO with the nature of dependency and distance covariance is well written.
4.	The numerical results are abundant, including tabular and image data. The experiments on image data illustrate the scalability of the proposed algorithms, unlike common experiments only on small-scale tabular datasets.

**Weaknesses:**

1.	The intuitive explanation for the distance covariance and its existing usages in statistics are not well explained.
2.	There are some missing references for fairness interventions, which I encourage the authors to include and compare, for example,
a.	Lowy, A., Baharlouei, S., Pavan, R., Razaviyayn, M. and Beirami, A., 2021. A stochastic optimization framework for fair risk minimization. arXiv preprint arXiv:2102.12586.
b.	Alghamdi, W., Hsu, H., Jeong, H., Wang, H., Michalak, P., Asoodeh, S. and Calmon, F., 2022. Beyond Adult and COMPAS: Fair multi-class prediction via information projection. Advances in Neural Information Processing Systems, 35, pp.38747-38760.
3.	Despite that the distance covariance is interesting, the reason why it is potentially a better metrics than other information-theoretic quantities such as mutual information and the Renyi maximal correlation is unclear to me. Distance covariance, MI and the maximal correlation are all zero when two random variables are independent; however, the maximal correlation satisfies Renyi’s 6 postulates for a good measure of dependency. It is encouraged that the authors spend more space to discuss the pros and cons regarding the dependency metrics, give illustrations on why one is better than the other and hopefully provide a simple numerical example.
4.	In the experimental results (Table 1 and 2), it seems that the proposed results consistently have higher accuracy and lower fairness violation. However, the proposed result is not too different from other methods as most of them are in the Lagrangian form, i.e., CE loss plus fairness/ independence constrains. It is encouraged that the authors explain clearly why the proposed method could lead to a consistently better acc-fariness trade-off point than other methods.

**Questions:**

Please refer to Weakness. I will consider raising the scores after the rebuttal period if the authors could address the weaknesses.

---

> ### Author Response · Authors · 2023-11-16
> **FAIR ATTRIBUTE CLASSIFICATION VIA DISTANCE COVARIANCE**
>
> **1. The intuitive explanation for the distance covariance and its existing usages in statistics are not well explained.**
>
> **Answer**: Thanks for your comment. In the submitted version, the related works section highlights the computational disadvantages of Hirschfeld-Gebelein-Rényi (HGR) maximal correlation and mutual information (MI). In this revised version, we  add discussions about advantages and disadvantages of the distance covariance, HGR and MI in the introduction section.
>
> Two random vectors (variables) are independent if and only if any of their HGR maximal correlation, MI, and distance covariance (DC) obtain a value of 0. Although all the above three can be used to characterize the independence of two random variables （vectors）, the DC **stands out** in terms of its computational efficiency and suitability for optimization.
>
>  * The definition of HGR of two random vectors is
> $$HGR(U,V)=\sup_{f,g}\ \langle f(U),g(V)\rangle.$$
> In practical computations, it is **IMPOSSIBLE** to traverse all $f,g$. Some classical methods are to approximate HGR by requiring $f,g$ belonging to linear space or Reproducing Kernel Hilbert Spaces. In [1], the authors solve it through a kernel approximation.
>
> * MI terms are **difficult** to estimate and optimize [2]. In [2], they replace both mutual information of the objective function and the function in constraint by their lower bound and upper bounds respectively.
>
> * For DC,
>
> 1. Empirical DC, as a statistic, can be **directly** computed from the samples, although population DC [3] is also challenge for computing since it requires knowledge of the analytical form of the distribution function and involves integration.
>
> 2. The empirical DC of the predicted target and sensitive attribute is **continuously differentiable and biconvex** when we treat the predicted target as a free variable. Furthermore, it can be expressed in matrix form, which makes it relatively more suitable for optimization and efficient for computation.
>
> [1]. J. Mary, C. Calauzenes, and N. E. Karoui. Fairness-Aware Learning for
> Continuous Attributes and Treatments. ICML, 4382–4391, 2019.
>
> [2]. J. Song, P. Kalluri, A. Grover, S. Zhao, and S. Ermon. Learning
> Controllable Fair Representations. AISTAT, 2164–2173, 2019.
>
> [3]. J. Liu, Z. Li, Y. Yao, F. Xu, X. Ma, M. Xu, and H. Tong. Fair Representation
> Learning: An Alternative to Mutual Information. SIGKDD, 1088–1097, 2022.
>
>
> **2. There are some missing references for fairness interventions, which I encourage the authors to include and compare**
>
> **Answer**: Thanks for your recommendation, we add them to related works and numerical experiment parts.

---

> ### Author Response · Authors · 2023-11-16
> **Rebuttal for the advantages of distance covariance**
>
> **3. Despite that the distance covariance is interesting, the reason why it is potentially a better metrics than other information-theoretic quantities such as mutual information and the Renyi maximal correlation is unclear to me.
> Distance covariance, MI and the maximal correlation are all zero when two random variables are independent; however, the maximal correlation satisfies Renyi’s postulates for a good measure of dependency.
> It is encouraged that the authors spend more space to discuss the pros and cons regarding the dependency metrics, give illustrations on why one is better than the other and hopefully provide a simple numerical example.**
>
> **Answer**: Thanks for your comment, we add this part to the introduction section.
>
> Although all the above three can be used to characterize the independence of two random variables (vectors), the distance covariance (DC) **stands out** in terms of its computational efficiency and suitability for optimization.
>
>  * For DC, Empirical DC, as a statistic, can be *directly* computed from the samples, although population DC [1] is also challenge for computing since it requires knowledge of the analytical form of the distribution function and involves integration.
>
> - In addition, the empirical DC is a **continuously differentiable and biconvex** function about the predicted target matrix  and the sensitive attribute matrix. In the fairness classification problem, the sensitive attribute matrix is known, so the empirical DC is a continuously differentiable convex function of the predicted target matrix, which is an elegant property for optimization.
>
> - Moreover, the paper [1] also demonstrates the advantages of DC by comparing with methods based on mutual information [2,3], demonstrating its advantages. We further utilized empirical distance covariance, not only reducing training time but also improving both accuracy and fairness.
>
> * For HGR and MI, we can only use the **estimations or upper/lower bounds** instead, which are used to approximate independence.
> In practice, this inevitably introduces biases and errors, subsequently reducing the utility of downstream classification tasks.
>
> - The definition of HGR of two random vectors is
> $$HGR(U,V)=\sup_{f,g}\ \langle f(U),g(V)\rangle.$$
> In practical computations, it is **IMPOSSIBLE** to traverse all $f,g$. Some classical methods are to approximate HGR by requiring $f,g$ belonging to linear space or Reproducing Kernel Hilbert Spaces. In [4], the authors use an $m$-out-of-$n$ bootstrap to estimate the singular values of a stochastic matrix from a finite sample and solve it through a kernel approximation.
>
> - MI terms are **DIFFICULT** to estimate and optimize, researchers try to replace the MI by the lower/upper bounds [5] or some variational methods [6].
>
> [1]. Liu et al. Fair Representation
> Learning: An Alternative to Mutual Information. SIGKDD, 1088–1097, 2022.
>
> [2] Creager et al. Flexibly fair representation learning by disentanglement. ICLR, 1436-1445, 2019.
>
> [3] Louizos et al. The variational fair autoencoder. arXiv preprint arXiv:1511.00830, 2015.
>
> [4]. Mary et al. Fairness-Aware Learning for
> Continuous Attributes and Treatments. ICML, 4382–4391, 2019.
>
> [5]. Song et al. Learning
> Controllable Fair Representations. AISTAT, 2164–2173, 2019.
>
> [6] J. Song, and E. Stefano. Understanding the Limitations of Variational Mutual Information Estimators. ICLR, 2019.
>
> **4.  In the experimental results (Table 1 and 2), it seems that the proposed results consistently have higher accuracy and lower fairness violation. However, the proposed result is not too different from other methods as most of them are in the Lagrangian form, i.e., CE loss plus fairness/ independence constrains. It is encouraged that the authors explain clearly why the proposed method could lead to a consistently better acc-fariness trade-off point than other methods.**
>
> **Answer**: Thanks for your comment. A consistently better result is from a **better model $+$ better model parameter selection $+$ better optimization**. As stated in the above, our DC regularization is a good choice for fair classification.
>
> In our optimization problems, there is a crucial parameter that plays a significant role in achieving optimal performance. The selection and adjustment of this parameter greatly influence the optimization process and the quality of the resulting solution. To address this, we employ the Lagrangian dual method as an alternative approach to update the balanced parameter and network parameters.
>
> In our experiments, we initially attempted to manually choose the balanced parameter. However, we found that the performance was slightly lower compared to using the Lagrangian dual method. This highlights the advantage of employing an adaptive approach.
>
> *Hope our reply can solve your concern. We are appreciated if you can change your decision and give us more score.*

---

> > ### Comment · Reviewer_8JtY · 2023-11-22
> >
> > I appreciate the authors for their response and additional explanations. I will stand my score as I think my concerns on the advantage of DC over the maximal correlation and MI are still not fully addressed. I agree that the computation of the maximal correlation and MI is hard; however, there are several surrogate (e.g., lower bounds, kernels, etc.) that allows the computation just like the empirical DC here. Also, the statistical properties between the empirical DC and population DC (isn't the empirical DC an estimate of DC?) is not very clear either.

---

> > > ### Author Response · Authors · 2023-11-23
> > > **Advantages of DC**
> > >
> > > Dear reviewer,
> > >
> > > Thank you for dedicating your time to reviewing our paper. We deeply appreciate your valuable feedback, as it will undoubtedly enhance the quality of our work.
> > >
> > > The empirical distance covariance serves as a **statistic** of the population distance covariance, employing finite samples.
> > >
> > > The discrepancy between the empirical DC and the population DC is attributed to **stochastic error**, which can be reduced by increasing the sample size. It is crucial to note that **the empirical DC almost surely converges to the population DC when the sample size tends to infinity**.
> > >
> > > For HGR and MI, they can use the estimations or upper/lower bounds instead. Apart from **stochastic error**, there are also **approximation errors** that arise due to the nature of the approximation itself. However,  increasing the sample size does not necessarily drop the approximation error.
> > >
> > > It is important to highlight that distance covariance serves as a **lower bound** for mutual information (MI), which is a desirable property established by Theorem 1 in [1]. Consequently, a smaller MI value implies a smaller distance covariance value, while the reverse is not necessarily true. Therefore, distance covariance is better than MI.
> > >
> > > [1]. Liu et al. Fair Representation Learning: An Alternative to Mutual Information. SIGKDD, 2022.

---

### Meta-Review · Area_Chair_TrMN · 2023-12-10

**Metareview:**

The paper introduces distance covariance, which is used to approximate statistical dependencies between the sensitive feature and predictions in fair classification. The authors provide consistency and sample complexity bounds for the distance covariance, and empirically evaluate over several tabular and image datasets. However, the reviewers have pointed out several issues of the proposed paper, which in my opinion (also shared by the reviewers) have not been convincingly addressed. In my opinion the main issues that remain are the following:

i) The claims regarding demographics parity and equality of opportunity seem to be in disagreement with prior literature stating that except for trivial cases, these two fairness notions cannot be achieved jointly. The rebuttal on this point remains unclear.  This is an important point that should be clarified in the revised version of the paper.


ii) The authors claim that  DC stands out in terms of its computational efficiency and suitability for optimization, which is to some extent correct. However, given that in fairness-performance trade-off the results are comparable to the ones provided by other methods, I find the the results on the computational efficiency, which were shared by the authors during the rebuttal and only on the adult data, insufficient for making a strong argument in favor of their approach. Regarding suitability for optimization, the proposed method requires the classifier to be differentiable, however, in the tabular domain, differentiable classifiers (e.g., neural networks) are often outperformed by non-differentiable ones. As a consequence, while I in general like the idea of distance covariance, the novelty and practical contributions of the paper seem still limited to me (given the vast amount of works to ensure demographic parity in fair classification).

In summary, agree with the reviewers that the authors should make a clearer case of the theoretical and practical advantages of their approach. I thus encourage the authors to improve their work based feedback, rebuttal and discussion with the reviewers.

**Justification For Why Not Higher Score:**

In its current form, I do not think that the paper provides significant contributions to the fair classification literature. Moreover, the paper requires major revision, and thus an additional complete round of reviews before being ready for publication.

**Justification For Why Not Lower Score:**

N/A

---

### Decision · Program_Chairs · 2024-01-16

Reject